# Gap-enhanced Raman tags for physically unclonable anticounterfeiting labels

Yuqing Gu[1,6], Chang He[1,6], Yuqing Zhang[1,2], Li Lin [1], Benjamin David Thackray[1] & Jian Ye [1,3,4,5]*

Anticounterfeiting labels based on physical unclonable functions (PUFs), as one of the powerful tools against counterfeiting, are easy to generate but difficult to duplicate due to inherent randomness. Gap-enhanced Raman tags (GERTs) with embedded Raman reporters show strong intensity enhancement and ultra-high photostability suitable for fast and repeated readout of PUF labels. Herein, we demonstrate a PUF label fabricated by drop-casting aqueous GERTs, high-speed read using a confocal Raman system, digitized through coarse-grained coding methods, and authenticated via pixel-by-pixel comparison. A three-dimensional encoding capacity of over $3 \times 10^{15051}$ can be achieved for the labels composed of ten types of GERTs with a mapping resolution of 2500 pixels and quaternary encoding of Raman intensity levels at each pixel. Authentication experiments have ensured the robustness and security of the PUF system, and the practical viability is demonstrated. Such PUF labels could provide a potential platform to realize unbreakable anticounterfeiting.

[1] State Key Laboratory of Oncogenes and Related Genes, School of Biomedical Engineering, Shanghai Jiao Tong University, Shanghai 200030, People's Republic of China. [2] School of Automation, Hangzhou Dianzi University, Hangzhou 310018, People's Republic of China. [3] Shanghai Key Laboratory of Gynecologic Oncology, Ren Ji Hospital, School of Medicine, Shanghai Jiao Tong University, Shanghai 200127, China. [4] Department of Nuclear Medicine, Ruijin Hospital, School of Medicine, Shanghai Jiao Tong University, Shanghai 200025, China. [5] Shanghai Med-X Engineering Research Center, School of Biomedical Engineering, Shanghai Jiao Tong University, Shanghai 200030, People's Republic of China. [6] These authors contributed equally: Yuqing Gu, Chang He. *email: yejian78@sjtu.edu.cn

Counterfeiting is a growing challenge worldwide, affecting a wide range of products from luxury articles to common consumer goods[1,2]. Counterfeiting does not just cause significant economic losses and social problems by undermining intellectual property law[2,3], but also threatens human health with fake pharmaceuticals[4–6]. Anticounterfeiting labels are among the most common solutions, such as holograms[7], watermarks[7], graphical barcodes[8], and security inks[8]. In addition, nanostructured-surface labels containing probes with surface-enhanced Raman scattering (SERS) signatures are modern extensions of conventional anticounterfeiting system[8], including Raman barcoding[9,10] and Raman patterning[11]. More advanced authentication labels have been developed based on molecular tags, e.g., DNA[12,13], peptides[14,15], and polymers[16], which encode information mainly through the sequence of their building blocks. However, deterministic processes are used to produce these labels[8], which can also be used for forgery. This is a fundamental insecurity of deterministic anticounterfeiting, whose security relies principally on technical barriers and limited access to materials for their fabrication.

Authentication systems based on physical unclonable functions (PUFs) have been recently developed for unforgeable anticounterfeiting[8]. PUFs, also called physical one-way functions, refer to physical objects with inherent, unique, and fingerprint-like features[8]. PUF labels, fabricated using stochastic processes, are random patterns constructed by disordered distributions of micro- or even nano-structures. The theoretical maximum number of unique PUF labels such a process can produce is known as the encoding capacity[17,18], which is fundamental to the effectiveness of PUF systems—the lower the encoding capacity, the easier the system to be cracked[8]. Also, false-positive rates play a significant role in practical applications of PUF labels. The encoding capacity must be sufficiently large so that the chance of generating two identical labels using the stochastic process is vanishingly small. If no deterministic method exists, which can mimic the stochastic results, the PUF approach is fundamentally secure. Thus, a good PUF label must be identifiable, unpredictable and physically unclonable[8,19]. A unique PUF label can be assigned to a single product, whose identity is then secure during the supply chain. PUF labels typically fall into two categories in terms of the authentication approach. The first relies on the point-by-point comparison of digitized patterns[8]. Kim et al.[20] reported an anticounterfeiting tag featuring randomly distributed nanowires coated with fluorescent dyes, and recorded the location and the color of nanowires to realize identification. The second depends on pattern recognition[8]. For example, Bae et al.[21] designed a fingerprint-like wrinkled microparticle as a PUF tag, whose minutiae including ridge ending and bifurcation were extracted as their unique features. Compared with pattern recognition, the point-by-point comparison takes advantage of a shorter read time and a lower level of false positives, despite the dependence on pattern orientation for identification, which can be circumvented by using alignment techniques.

Chemical methods have been extensively adopted to produce anticounterfeiting tags, such as luminescent nanomaterial-based security inks[22], spectrally distinct upconversion nanocrystal-based particle barcodes[23], nucleic acid[24,25] or peptide[26] based molecular tags, and so on. Such chemically fabricated PUF labels have many advantages[17,20,21,27–29]. Firstly, chemical approaches are stochastic processes, as required for PUFs. Secondly, chemically generated PUF labels usually have large encoding capacities due to the randomness and large parameter space offered by solution chemistry[8], which can be further improved by adding more tags or choosing tags with multiple detectable chemical characteristics. Thirdly, chemical fabrication of PUF labels does not usually require costly equipment, making it more convenient, economical, and suitable for mass production.

Optical PUF labels[30–32] can be chemically produced, with their readout based on optical responses such as fluorescence or Raman scattering from tags. Optical readout is a non-contact, fast, and relatively convenient technique, enabling high-encoding capacity using multiple spectral features. Among optical techniques, surface-enhanced Raman spectroscopy (SERS) is promising for PUF systems[33,34]. Upon laser irradiation, SERS tags produce fingerprint Raman spectra unique to the reporter molecule used. Compared with scattering-based PUF labels, SERS-based ones can have much more responses at each pixel of the read pattern owing to distinguishable spectra from various types of tags. In addition, much narrower spectral linewidth of SERS tags than fluorophores greatly alleviates cross-talk between neighboring peaks[35], allowing a larger encoding capacity. Furthermore, demultiplexing methods such as the classical least squares (CLS) method based on spectral profile recognition enable even a larger encoding capacity in contrast to the conventional Raman band recognition. Also, SERS tags tend to be more material-stable and photostable than conventional fluorophores[36], though some lanthanide-based luminescent materials have been reported to show a quite high photostability[17].

Herein we demonstrate fabrication and authentication of SERS-based PUF labels for anticounterfeiting. The labels are stochastic two-dimensional (2D) patterns formed by drop-casting various types of core-shell SERS nanoparticles (NPs), known as gap-enhanced Raman tags (GERTs)[37–42], on a silica substrate (Fig. 1a). The disordered distribution of NPs at the nanoscale and the randomness of the pattern they produce make the labels impossible to counterfeit. The PUF labels are read using a lab-scale confocal Raman system by performing Raman mappings with different resolutions. The readout signals are then digitized based on both Raman spectral profiles (extracted via CLS) and Raman intensity levels at each pixel. The theoretical encoding capacity of PUF labels constructed this way can be calculated as $L^{n \times m}$, where $L$, $n$, and $m$ represent the number of Raman intensity levels per pixel, number of NP types, and number of pixels, respectively (see Fig. 1b). Therefore, the PUF label can be regarded as three-dimensionally (3D) encoded. The encoding capacity increases exponentially with the increase of $n$ and $m$, and as a power function of $L$. In this work, we have demonstrated a remarkable theoretical encoding capacity exceeding $3 \times 10^{15051}$ using a PUF label composed of ten types of SERS NPs with a mapping resolution of $50 \times 50$ pixels and quaternary encoding of Raman intensity levels at each pixel. Such a large encoding capacity ensures the impossibility of duplicating. In addition, authentication experiments have indicated that there is an obvious distinction between the similarity index ($I$) of the same PUF labels and that of different ones, further favoring the practical applications of such labels.

## Results

**Synthesis and characterization of GERTs**. There are various metallic NPs, including Au nanospheres[43], nanorods[44], nanotriangles[45], and nanoflowers[46] that can be used as SERS tags for PUF labels. Raman reporter molecules are typically adsorbed on the surface of metallic NPs to produce these SERS tags and are mainly enhanced by an electromagnetic effect produced by the localized surface plasmon resonance (LSPR) of the NPs. Consequently, their SERS signals are easily affected by the NP states and the environmental conditions (e.g., oxygen, moisture). Herein we use core-shell structured GERTs to construct our PUF labels. GERTs are typically composed of a Au core and shell with Raman reporters embedded in a nanometer gap between them (Fig. 2a). GERTs were

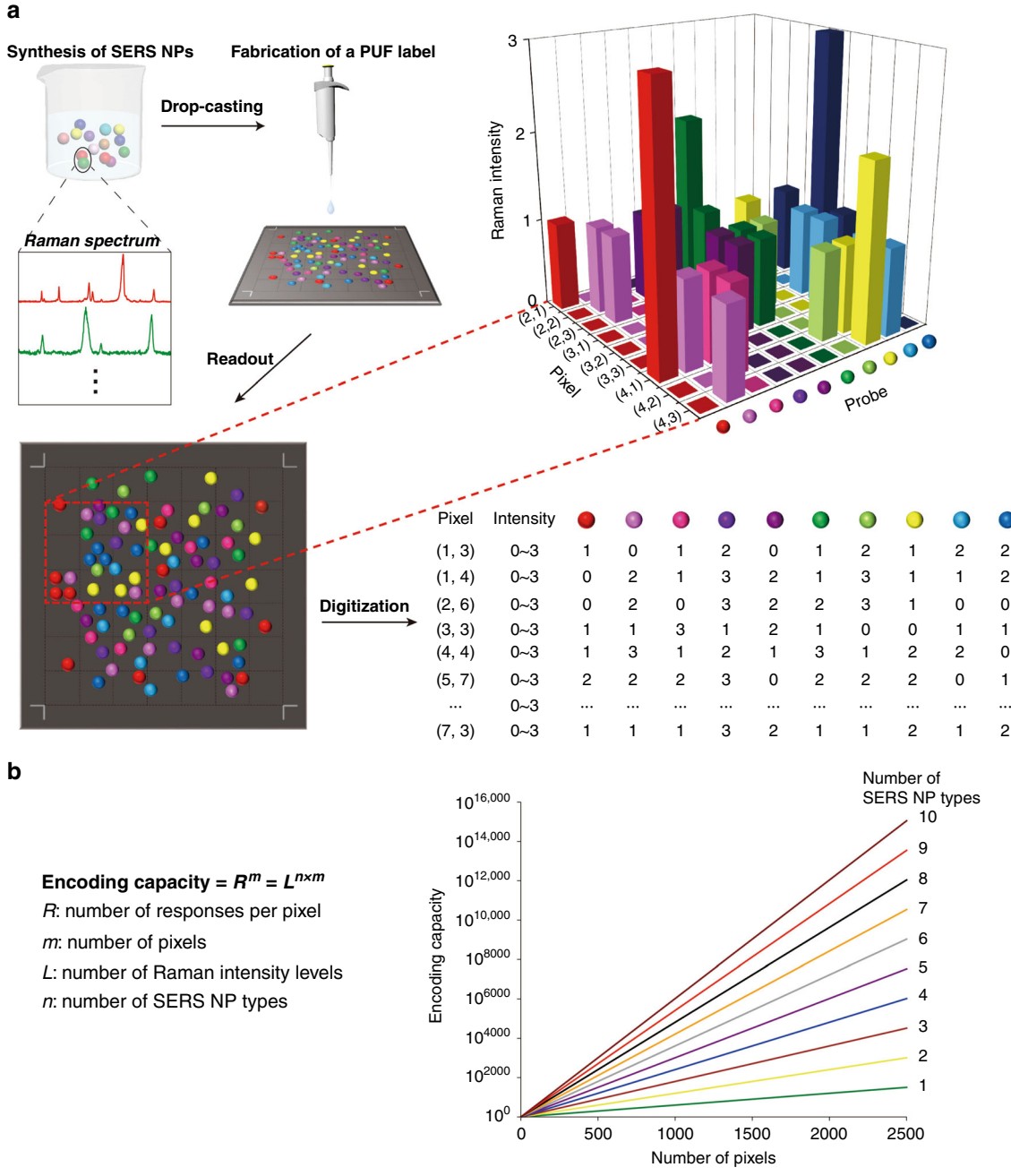

**Fig. 1 Fabrication and encoding of Raman PUF labels. a** A Raman signal-based physical unclonable function (PUF) label can be fabricated in four steps. First, surface-enhanced Raman scattering (SERS) nanoparticles (NPs) functionalized with ten different Raman reporter molecules are synthesized by a wet chemistry method. Each type of SERS NPs emits a unique vibrational Raman spectrum. Second, SERS NPs are deposited on a substrate and stochastically form a two-dimensional (2D) pattern after drying. The PUF label can then be read using a confocal Raman system by mapping and recording the signals at different points (e.g., 8 × 8 pixels in this example). The various SERS NPs at different locations on the substrate produce Raman spectra with specific bands and intensities. A PUF label is finally produced by digitizing Raman mapping signals at each pixel in terms of the type of SERS NPs and the level of Raman intensities with the combination of the physical position of each pixel. **b** Calculation of the encoding capacity of a Raman PUF label. The formula (left) and the graph (right) show that the encoding capacity of a PUF label increases exponentially with the number of pixels ($m$) and as a power function of the number of possible responses for each pixel ($R$) in the recorded pattern. Furthermore, the number of responses of each pixel is a power function of the number of Raman intensity levels ($L$) and an exponential function of the type of SERS NPs ($n$). For example, if $n = 10$, $L = 4$, and $m = 2500$, the encoding capacity will be $4^{10 \times 2500}$ ($3.2 \times 10^{15051}$). This figure composition refers to the ref. [7].

synthesized following a previously reported method[47], using uniform reporter-decorated Au NPs as seeds to grow the Au shell by reduction of $Au^+$. We selected ten types of thiolated aromatic molecules as Raman reporters to prepare GERTs, namely 1,4-benzenedithiol (1,4-BDT), 4-nitrobenzenethiol (4-NBT), 4-methylbenzenethiol (4-MBT), 2-mercapto-5-nitrobenzimidazole

(2-M-5-NBI), 2-mercapto-6-nitrobenzothiazole (2-M-6-NBT), 2-nitrobenzenthiol (2-NBT), 2-chlorobenzenethiol (2-CBT), 4-chlorobenzenethiol (4-CBT), 4,4'-biphenyldithiol (4,4'-BPDT), and 2-naphthalenthiol (2-NT) (see molecular structures in Fig. 2b). We refer to the GERTs embedded with 1,4-BDT molecules as 1,4-BDT GERTs, and abbreviate all others in the same fashion. All reporter

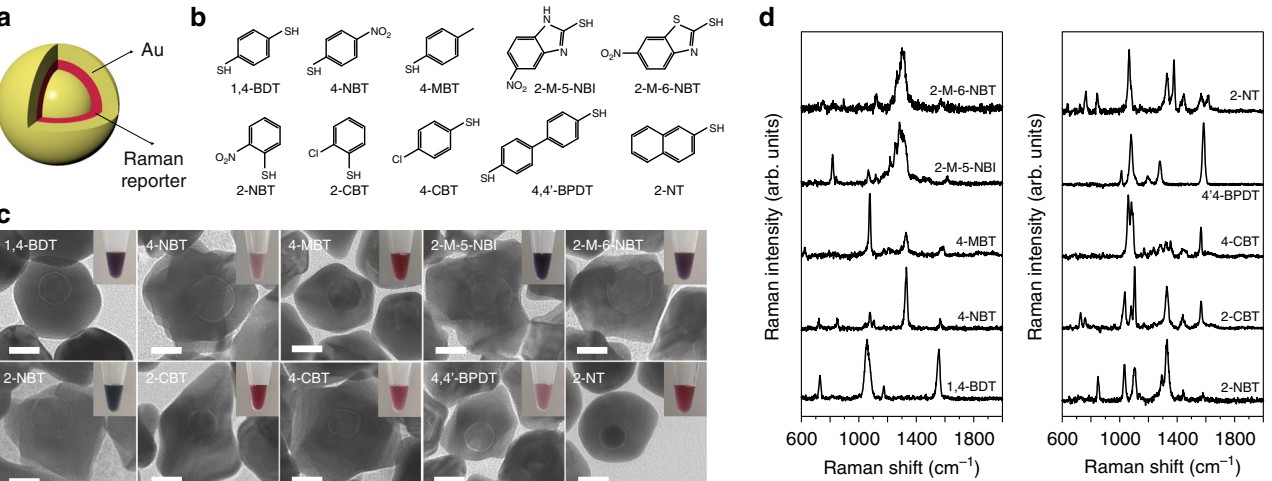

**Fig. 2 SERS NPs with embedded Raman reporters, termed as gap-enhanced Raman tags (GERTs), used for fabricating Raman PUF labels. a** Schematic representation of a GERT. **b** Molecular structures, **c** transmission electron microscopic (TEM) images, and **d** SERS spectra of ten GERTs with different embedded reporter molecules. Insets in **c** are the corresponding photos of various GERT solutions. All scale bars are 20 nm.

molecules have thiol groups that can form robust covalent bonds with Au. The representative transmission electron microscopy (TEM) images indicate a distinct nanogap of ~1 nm between the metallic core and shell for all GERTs, with a NP size of $53.4 \pm 2.7$, $69.8 \pm 10.0$, $49.2 \pm 3.3$, $68.7 \pm 6.0$, $62.8 \pm 2.9$, $79.9 \pm 6.2$, $44.0 \pm 3.5$, $70.7 \pm 6.9$, $46.5 \pm 3.3$, and $49.7 \pm 3.2$ nm (from left to right, top to bottom in Fig. 2c), respectively. Although some GERTs show an irregular shape of the external shell, they are uniformly dispersed without aggregation (Fig. 2c and Supplementary Fig. 1). The extinction spectra indicate that all GERTs exhibit a single pronounced resonance peak in the visible range from 500 to 650 nm (Supplementary Fig. 2), behaving like homogeneous solid Au NPs. The insets of Fig. 2c present the typical colors of the corresponding aqueous GERTs: purple, pink, ruby, blue, purple, blue, ruby, red, pink, and ruby (from left to right, top to bottom), respectively. All GERTs exhibit strong Raman signals when excited by a 785 nm laser (Fig. 2d) and the narrow linewidths of their vibrational Raman bands (see detailed mode assignments in Supplementary Table 1) along with their unique spectral profiles allows use of demultiplexing methods (e.g., CLS) to obtain a large encoding capacity.

The off-resonantly excited GERTs (with a localized surface plasmon resonance in the visible range but excited by 785 nm near-infrared laser) show large Raman enhancement due to a combination of electromagnetic field enhancement and electron transport effect across molecular layer in the nanogaps[41,48], and therefore lead to a number of important properties favorable for PUF labels: (1) large enhancement factor, detectable down to a single-NP level[42,49], leading to the fastest (to the best of our knowledge) readout speed with a good signal-to-noise ratio; (2) ultra-photostability under repeated readout due to the off-resonance excitation condition[37,50,51], leading to excellent reproducibility; (3) ultra-stable material properties in various environments (for example, humid environment), resulting in easy storage and a long shelf-life of prepared labels[37]; (4) suitability for NIR laser excitation, resulting in low Raman background from the PUF substrate or package materials. We have to emphasize that the conventional plasmonic dimers or aggregates are inappropriate for the PUF labels since the SERS hot spots from them are apt to photobleaching[37,52].

**Fabrication and digitization of PUF labels composed of one-type of GERTs.** A PUF label consisting of one-type of GERTs was first fabricated by drop-casting 2 µL of 4-NBT GERTs solution (0.6 nM) on a silica substrate. The GERTs are randomly distributed on the substrate and form a stochastic pattern at a nanoscale level (Fig. 3a, b), which is almost impossible to reproduce by chance. An area of $100 \times 100 \, \mu m^2$ (Fig. 3a) was selected as a PUF label and some metallic marks (indicated by red arrows in Fig. 3a and Supplementary Fig. 3) were created for easy location during future repeat measurements. The pattern of the scanning electron microscopy (SEM) image of the label coincides well with that of the bright-field image. The magnified section of the image shows various distribution states of the GERTs on the substrate, for example, small and large aggregations, trimers, and even single NPs (Fig. 3b). The complexity and uniqueness of the distribution of GERTs at the nanoscale level ensures that the random pattern cannot be copied, either by chance or with any deterministic method. The PUF label was then read by performing Raman mapping with a resolution of $2 \times 2$, $10 \times 10$, and $50 \times 50$ pixels using a confocal Raman system (785 nm laser, $3 \times 10^5 \, W \, cm^{-2}$, 60× objective lens, an exposure time of 10 ms per pixel) to acquire the averaged Raman signal of the GERTs at each pixel. The obtained Raman mapping images plotted using the Raman band at 1078 $cm^{-1}$ for 4-NBT GERTs show the patterns, where intensity of each pixel varies widely and is mainly determined by the number of GERTs in each pixel (Fig. 3c). Increasing the number of pixels adds to the complexity of the mapping pattern. It is noteworthy that the difference in Raman intensity between the maximum and the minimum values of the measured pixels increases with higher resolution (Fig. 3c, d). This is because at lower resolution each pixel contains more NPs and the averaged Raman intensity of each pixel is therefore closer to the mean intensity of all GERTs within the whole mapping, which means a smaller difference among pixels. In addition, the Raman spectral profile is less like that of pure 4-NBT GERTs with a lower resolution (point 1 and 2 in Fig. 3d), possibly because the selected pixels consist of a larger percentage of empty area, which no NPs occupy, resulting in a more significant signal contribution from the substrate.

We first consider the encoding capacity of the PUF label realized in two dimensions: the first dimension is the physical position of pixels, and the second is the Raman intensity at 1078 $cm^{-1}$ of 4-NBT GERTs. The above Raman mapping images with different resolutions were digitized as follows. First, the raw Raman signals (intensity at 1078 $cm^{-1}$) were processed with a

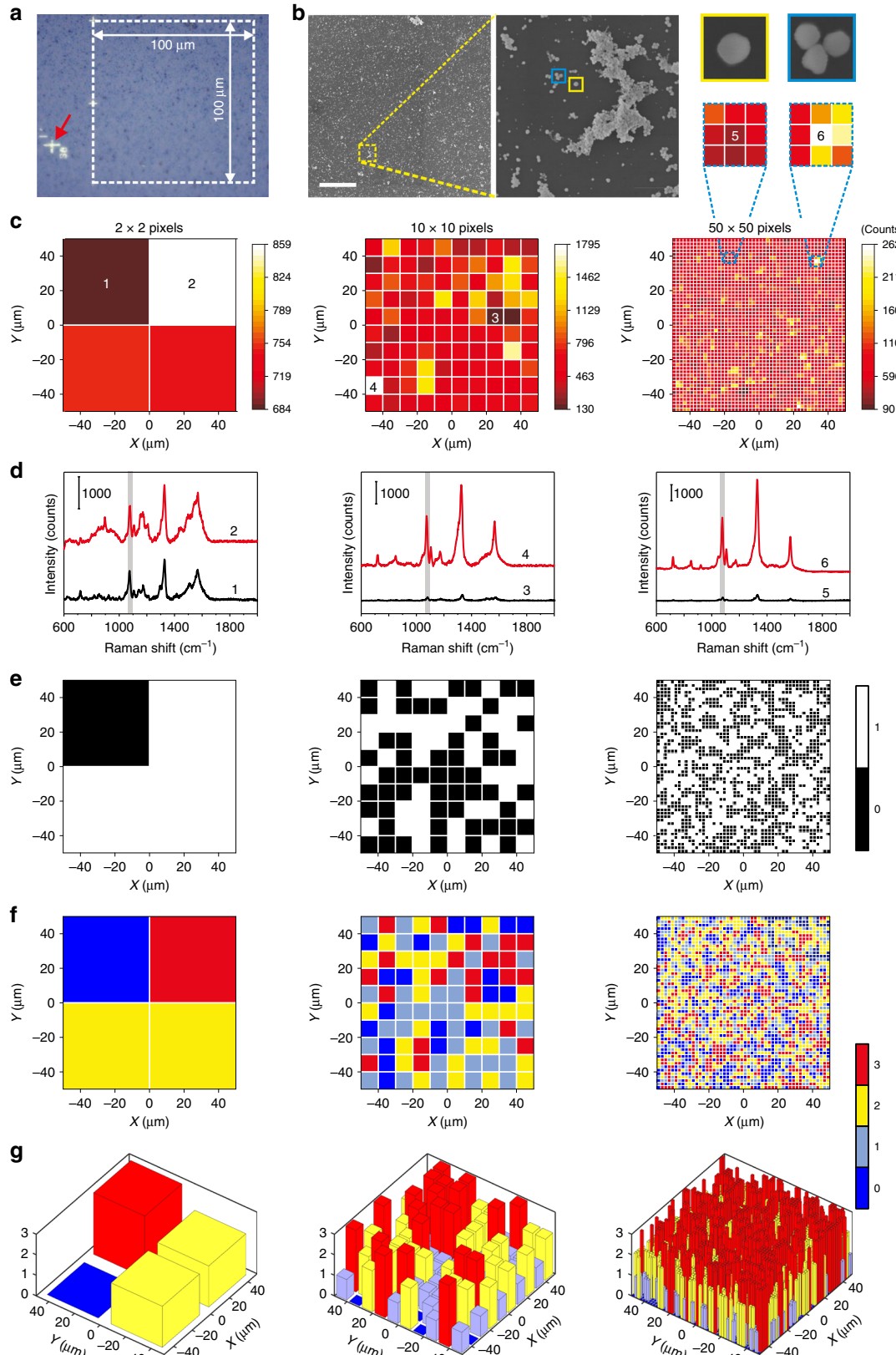

**Fig. 3 Fabrication and digitization of a PUF label composed of 4-NBT GERTs. a** A bright-field image and **b** a scanning electron microscopy image (corresponds to the dashed square area in **a**) of a PUF label composed of 4-NBT GERTs in an area of 100 × 100 μm² on a SiO₂ substrate. Scale bars are 20 (left) and 1 μm (right). **c** Readout of the PUF label by the Raman mapping (plotted using the band at 1078 cm⁻¹) and the corresponding digitization of **e** binary encoding and **f**, **g** quaternary encoding of Raman intensity levels at each pixel with a resolution of 2 × 2 (left), 10 × 10 (middle), and 50 × 50 (right) pixels. Panel **d** shows the SERS spectra at the points (1–6) indicated in **c**. Panels **f** and **g** show the two-dimensional (2D) and three-dimensional (3D) plots of digitization of the PUF label based on quaternary encoding of Raman intensity levels, respectively.

method of $Z$-score. In statistics, the $Z$-score is the signed fractional number of standard deviations by which the value of an observation or data point is above the mean value of what is being observed or measured[53]. With this $Z$-score method, the raw readout matrices were standardized to a set with the mean of 0 and the standard deviation (SD) of 1 for both binary and quaternary encoding. The latter achieves a higher spatial density of encoding capacity by increasing the number of possible responses from each pixel. Next, a global search (GS) algorithm was employed for threshold determination, after which the standardized mapping data were quantized to a value from the set {0, 1} (for binary encoding) or {0, 1, 2, 3} (for quaternary encoding) using a coarse-grained coding method (see more details in Methods). Figure 3e, f show the two-dimensional (2D) plot of the digitization results of binary and quaternary encoding of the Raman intensity levels in each pixel with different resolutions, respectively. In order to present the digitized level of each pixel more explicitly, a three-dimensional (3D) version of Fig. 3f was plotted (see Fig. 3g). For convenience, we have

recorded the digital label as a matrix in the format of $\begin{bmatrix} a_1 \\ \vdots \\ a_m \end{bmatrix}$,

where $m$ represents the number of pixels. For instance, the digital mapping with a resolution of $2 \times 2$ pixels (left columns in Figs. 3e, f)

could be recorded as $\begin{bmatrix} 0 \\ 1 \\ 1 \\ 1 \end{bmatrix}$ for binary coding and $\begin{bmatrix} 0 \\ 3 \\ 2 \\ 2 \end{bmatrix}$ for

quaternary coding. Matrices for digitization of high-resolution Raman mapping images ($10 \times 10$ and $50 \times 50$ pixels) can be found in the Supplementary Materials (Supplementary Fig. 4). For binary encoding, each pixel has two responses (0 or 1). Therefore, the theoretical encoding capacity of a PUF label with a resolution of $50 \times 50$ pixels is $2^{2500}$ ($3.8 \times 10^{752}$). An even larger encoding capacity of $4^{2500}$ ($1.4 \times 10^{1505}$) could be achieved with quaternary encoding using a PUF label ($50 \times 50$ pixels) with four responses (0, 1, 2 or 3) per pixel. In real conditions, the encoding capacity of the PUF label would shrink if an error margin is introduced for authentication, and the shrinkage has relations with pattern distribution. The digitized mappings in Fig. 3e, f show uniform distributions of different intensity levels, especially the ones with higher resolutions, which is in favor of a large real encoding capacity.

**PUF labels composed of multi-types of GERTs.** To further enlarge the encoding capacity, we added another dimension[17] for encoding: the type of SERS NPs, thus obtaining a 3D encoded PUF label. It is known that the number of responses per pixel increases exponentially with the number of types of GERTs used in label fabrication. The unique spectral profile and the ultra-narrow linewidth of Raman bands offer almost unlimited encoding capacity in contrast to fluorescent dyes. PUF labels with an area of $100 \times 100 \, \mu m^2$ consisting of three (Supplementary Figs. 5 and 6) or ten (Fig. 4 and Supplementary Figs. 7–9) types of GERTs were fabricated to verify its feasibility. Herein, a ten-type GERT PUF label is demonstrated as an example, which can be created by following a similar procedure described above. Specifically, 2 μL of the mixed aqueous solution containing 1,4-BDT GERTs (0.2 nM), 4-NBT GERTs (0.2 nM), 2-NBT GERTs (0.2 nM), 2-M-5-NBI GERTs (0.3 nM), 2-M-6-NBT GERTs (0.3 nM), BPDT GERTs (5 nM), 4-MBT GERTs (20 nM), 2-CBT GERTs (20 nM), 4-CBT GERTs (20 nM) and 2-NT GERTs (20 nM) was drop-cast on a silica substrate. The concentrations of different GERTs vary because they have been adjusted to ensure that the Raman intensity of each type of GERTs is on the same order of

magnitude for easier decoding. It can be seen in the bright-field image (Fig. 4a) and SEM images (Fig. 4b) that various GERTs form mixed multilayers on the substrate. Magnification of the SEM image shows that most NPs form aggregates, though there still exist single GERTs (Fig. 4b). The as-prepared PUF label was then read by Raman mapping ($10 \times 10$ pixels) with a slightly longer exposure time of 50 ms per pixel (785 nm laser, $3 \times 10^5$ W cm$^{-2}$, 60× objective lens) in order to acquire stronger signals for easier decoding. Since the Raman spectra of these ten types of GERTs overlap in the range of 802–1600 cm$^{-1}$ (gray area in Fig. 4c), the integral area of the Raman bands in this range was used to plot the 2D and the 3D mapping images to show the distribution of all GERTs in the PUF label, as presented in Fig. 4d, e, respectively. To digitize the data in terms of GERTs types, we employed a non-negative least squares (NNLS) method to demultiplex the measured Raman spectrum in each pixel. NNLS assumes that the experimental Raman spectrum is a linear superposition of the pure spectra of each type of GERTs. The weight of each component, which is prevented from being negative in NNLS, is determined through an optimization process to obtain a best-fit spectrum with the least difference from the measured spectrum (see more details in Methods). The measured and best-fit Raman spectra from an example pixel indicated by a dashed green box in Fig. 4d show good correspondence (Fig. 4c). Intensities of bands at 1058 cm$^{-1}$ for 1,4-BDT GERTs, 1078 cm$^{-1}$ for 4-NBT GERTs, 1077 cm$^{-1}$ for 4-MBT GERTs, 817 cm$^{-1}$ for 2-M-5-NBI GERTs, 1300 cm$^{-1}$ for 2-M-6-NBT GERTs, 1033 cm$^{-1}$ for 2-NBT GERTs, 1104 cm$^{-1}$ for 2-CBT GERTs, 1060 cm$^{-1}$ for 4-CBT GERTs, 1585 cm$^{-1}$ for 4,4'-BPDT GERTs, and 1066 cm$^{-1}$ for 2-NT GERTs were adopted to create Raman mapping images for each of the ten GERTs (Fig. 4f and Supplementary Fig. 8a). Among them, the intensities of 2-M-5-NBI GERTs, 2-M-6-NBT GERTs and 4,4'-BPDT GERTs are relatively low, which may be due to uneven distribution of the NPs. It should also be noted that all types of GERTs show the strongest signals at the same pixel (indicated by an arrow in Fig. 4f). It is unlikely that there are actually maximal numbers of all types of GERTs at this pixel. We speculate that it is more likely due to partial overlap of the Raman bands of different GERTs causing signal from a small number of exceptionally bright particles to bleed into other components during the NNLS demultiplexing method. This issue could be avoided by either choosing more spectrally distinct Raman reporters or by improvements to the demultiplexing algorithm.

The separated mapping images were then quantized with the Gerchberg–Saxton (GS) algorithm and the coarse-grained coding method. Figures 4g, h show the 2D (see 3D version in Supplementary Fig. 7b) digitized mappings of the PUF label with a resolution of $10 \times 10$ pixels for binary and quaternary encoding, respectively. Compared with the PUF label composed of only 4-NBT GERTs, the ten-type GERT label has an additional dimension for encoding—the GERT type. Therefore, the number of responses per pixel is increased to $2^{10}$ for binary encoding or $4^{10}$ for quaternary. The digitized pattern could be recorded as a

two-dimensional matrix $\begin{bmatrix} a_{11} & \cdots & a_{1n} \\ \vdots & \ddots & \vdots \\ a_{m1} & \cdots & a_{mn} \end{bmatrix}$, where $m$ stands for

the number of pixels and $n$ represents the type of GERTs. Herein, we have assigned $n$ as 1–10 for 1,4-BDT GERTs, 4-NBT GERTs, 4-MBT GERTs, 2-M-5-NBI GERTs, 2-M-6-NBT GERTs, 2-NBT GERTs, 2-CBT GERTs, 4-CBT GERTs, 4,4'-BPDT GERTs, and 2-NT GERTs, respectively. The matrices for digitization of Raman mapping images are presented in Supplementary Fig. 4. Compared to the one-type GERT PUF label, the ten-type label with a resolution of $10 \times 10$ pixels has a much larger theoretical

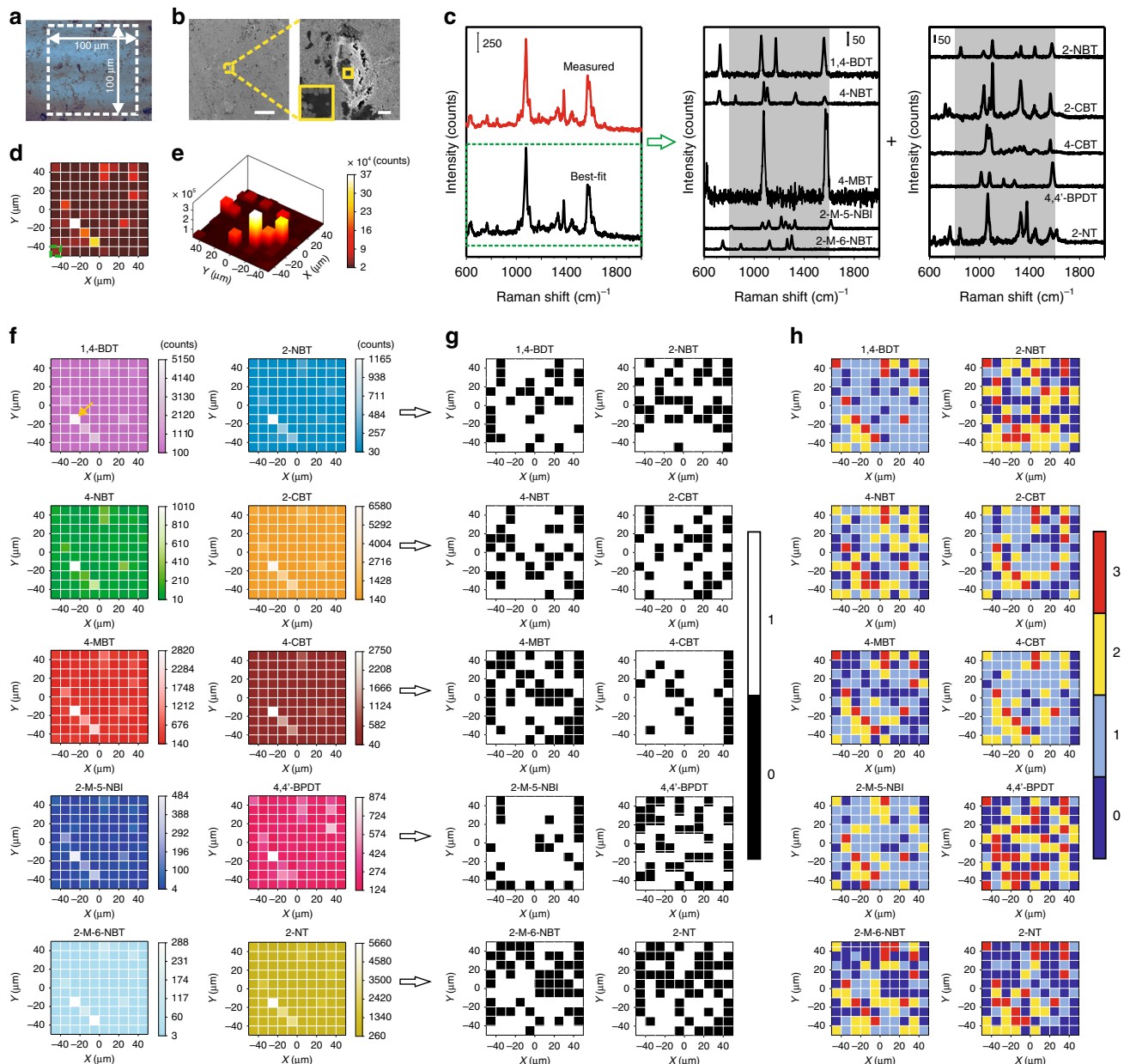

**Fig. 4 Fabrication and digitization of a PUF label composed of ten types of GERTs. a** A bright-field image and **b** the scanning electron microscopy image (corresponds to the dashed square area in **a** of a PUF label fabricated by ten types of GERTs in an area of $100 \times 100\ \mu m^2$ on a SiO$_2$ substrate. Scale bars are 20 (left) and 1 μm (right). **c** Spectral demultiplexing to quantify the abundance of ten types of GERTS at the position indicated by a green dashed box in panel **d**. The best-fit spectrum (black) is obtained by fitting the measured Raman spectrum (red) with ten pure reference Raman spectra using the non-negative least squares (NNLS) method. **d** 2D and **e** 3D plot for the readout of the PUF label by the Raman mapping (with a resolution of $10 \times 10$ pixels) of multiplexed measured Raman signals using the integrated areas of Raman bands from 802 to 1600 cm$^{-1}$ (as indicated in the gray area in panel **c**). **f** 2D plot for the readout of the PUF label by the Raman mapping (with a resolution of $10 \times 10$ pixels) of demultiplexed Raman signals and 2D plot of the corresponding digitizations using **g** binary encoding and **h** quaternary encoding of Raman intensity levels at each pixel.

encoding capacity of $(2^{10})^{100}$ $(1.1 \times 10^{301})$ and $(4^{10})^{100}$ $(1.1 \times 10^{602})$ for binary and quaternary encoding, respectively. When the imaging resolution is increased to $50 \times 50$ pixels (see Supplementary Figs. 8 and 9), the encoding capacity could be further expanded up to $(2^{10})^{2500}$ $(5.6 \times 10^{7525})$ and $(4^{10})^{2500}$ $(3.2 \times 10^{15051})$, over 6000 and 13,000 orders of magnitude larger than one-type GERT labels with the same resolution for binary and quaternary coding, respectively. While such a huge encoding capacity may appear somewhat redundant, it is important to note that in practice, no fabrication method will access all these states with equal probability, and small variations in readout can

significantly reduce the real capacity of any method[8]. Considering such real-world deviations from the idealized situation, a minimum encoding capacity of $10^{300}$ has been previously suggested[8,41]. Our method satisfies and exceeds this requirement, offering a significant additional hedge against further real-world imperfections in fabrication and readout that may not always be anticipated.

**Authentication of PUF labels.** In practical use, the PUF label needs to be authenticated repeatedly, requiring good reproducibility

between scans of the same label and a distinct difference between different labels. It is therefore important to quantify and study the similarity of readouts of the same label and different ones. Herein, we have introduced the similarity index ($I$) to evaluate the similarity of two digitized labels. It is found through comparison of two digitized images pixel by pixel[54], calculated by the percentage of zero in the matrix acquired from subtracting the matrix of one label from the other (see more mathematical details in Methods). To simulate the authentication process in real conditions, a total of 100 different PUF labels were selected for measurements. The readout of each label was repeated three times and the obtained matrices were standardized with the method of Z-score. Afterwards, a training set of 70 PUF labels was used for the search of common thresholds for digitization through the global optimization algorithm and the test set of the rest 30 labels was used to verify whether the thresholds are reasonable (see more details in Methods). Briefly, the most appropriate thresholds should result in not only relatively high reproducibility for the same labels but also significant disparity between different ones. The digitized matrices then went through pairwise comparison to get similarity indexes $I$ of the same PUF labels and those of different ones.

First, four labels (Supplementary Fig. 10) are chosen as examples to demonstrate the digitizing effects, as presented in Fig. 5a–d (see Supplementary Fig. 11 for 3D version), where all digital patterns show uniform distributions of various intensity levels. Herein, we use matrices with a resolution of $50 \times 50$ pixels for authentication because higher resolutions tend to give rise to fewer false positives[8]. It can be observed that the three digitized patterns from three measurements of the same label have a high degree of resemblance (Fig. 5a, b) and big similarity indexes ($I_{11'-1}$, $I_{11'-2}$, and $I_{11'-3}$) of around 94 and 84% for binary and quaternary encoding, respectively (Supplementary Table 2). The fact that the match does not reach 100% can most likely be explained by instability of the Raman system (including laser and optical alignment) and signal fluctuation from the SERS NPs. For example, instability of the laser power, variation of the photon detection efficiency of the charge-coupled device (CCD) induced by temperature variation, and shift in optical alignment[55] can exert pronounced negative effects on the reproducibility of PUF labels. In addition, our previous work has shown that the Raman signals of GERTs may fluctuate under continuous laser irradiation, probably induced by the reorientation and decomposition of Raman reporter molecules[50]. Table S2 shows that $I_{11'}$ for binary encoding is larger than that for quaternary encoding, which makes sense since the former has only two responses per pixel, half of the latter. It is therefore less likely that the same pixel would be quantified and digitized as a different Raman intensity level under repeat measurement for binary encoding, resulting in a higher reproducibility. When it comes to the digitization of additional three labels numbered 2, 3 and 4 (Fig. 5c, d), obvious disparity could be observed between these labels and the one displayed in Fig. 5a, b. Their similarity indexes ($I_{12}$, $I_{13}$, and $I_{14}$) have significantly dropped to around 57 and 30% for binary and quaternary encoding, respectively (Supplementary Table 2). Then, we randomly selected 10 PUF labels for validation between their first and second measurement to clearly show the robustness of the authentication algorithm, as presented in Fig. 5e, f of a $10 \times 10$ matrix for binary and quaternary encoding, respectively. The sharp contrast of the matrices indicates great disparity between $I$ of the same labels and that of different ones, which means that our PUF system can successfully fabricate unique labels. To demonstrate the validation results more completely, the distribution histograms of similarity indexes are plotted (Fig. 5g, h) with a total sample capacity of 300 and 21,735 for the same PUF labels and different ones. Both the training set and test set show that similarity indexes of the same labels are well separated

from those of different labels with a gap of around 20 and 30% for binary and quaternary coding, respectively, indicating that it is possible to distinguish real labels from the duplicate ones. According to the histograms, here we preliminarily suggest an error margin of 85 and 70% for binary and quaternary encoding, respectively, which could be further optimized with more sample data in real conditions. In practical use, a best digitization threshold (or list of three thresholds for quaternary encoding) would be found by the manufacturer through the optimum algorithm with sufficient sample capacity of PUF labels. This threshold is then applied for the digitization of all PUF labels. At the point of authentication in the supply chain, the PUF label is scanned and digitized, and the label data is sent to be compared to labels in the database at the cloud server. If a label is found in the database with a similarity index above the error margin, it is declared a match and the label is authenticated as genuine. Otherwise the label is rejected. The thresholds used for digitization and the error margin therefore play an important role in determining the security of the system and should be chosen carefully. During the authentication process, the most likely existing vulnerability is that the digitized signals uploaded to the cloud server may be replaced by an attacker, who can scan and get the digital signal of one genuine PUF label and repeatedly apply it to replacing uploaded signals of fake labels. However, we believe that the digitized data can be further algorithmically encoded and then sent to the cloud server, where the decoding and verification process is carried out followed by the feedback to the terminal device. With the help of an effective coding algorithm, which can be renewed frequently, the PUF system could effectively avoid direct transmission of digitized signals, thus solving the problem of data substitution by attackers.

**High-speed readout of PUF labels.** Typically, Raman mapping using the confocal system was performed at a relatively low speed. For example, readout of the above-mentioned PUF labels used the conventional STAGE mode, as shown in Supplementary Fig. 12, where the laser spot is fixed and the $X$-$Y$ stage is moving. With exposure time set to 10 ms per pixel, 8 s, 90 s, and 20 min were needed to obtain a Raman map with resolutions of $2 \times 2$, $10 \times 10$, and $50 \times 50$ pixels, respectively. It is worth noting that the exposure time accounts for a small fraction of the measurement duration. For example, the total exposure time of the one-type GERT PUF label with 2500 pixels is just 25 s (10 ms × 2500), about 2% of the total measurement time of 20 min. Most of the time is consumed by stage movements and data processing conducted pixel by pixel during the measurement. For practical use, faster readout is usually required. To solve the problem, a high-speed DuoScan mode (Fig. 6a) along with a SWIFT mode on the confocal Raman system was used[56]. To demonstrate high-speed readout in DuoScan mode, a PUF label ($100 \times 100 \ \mu m^2$) fabricated using 4-NBT GERTs was mapped with a shortened exposure time of around 0.7 ms per pixel, which is by now the shortest acquisition time of commercial electron-multiplying charge-coupled device (EMCCD) as far as we know[42]. Only 6 s was needed to acquire a Raman map with a resolution of $50 \times 50$ pixels (Fig. 6b), and good signal-to-noise ratios can be found at each pixel, which confirms that a valid readout of the PUF label could be achieved under the DuoScan/SWIFT mode. These dramatic improvements can be attributed to three factors. First, GERTs can produce really strong Raman signals, making it possible to reduce the acquisition time down to 0.7 ms per pixel. Second, Raman mapping is realized by rapid movement of the laser spot across the labels instead of mechanical movement of the stage, thus significantly shortening scanning time. The laser movement in $X$ and $Y$ directions is controlled by two galvo

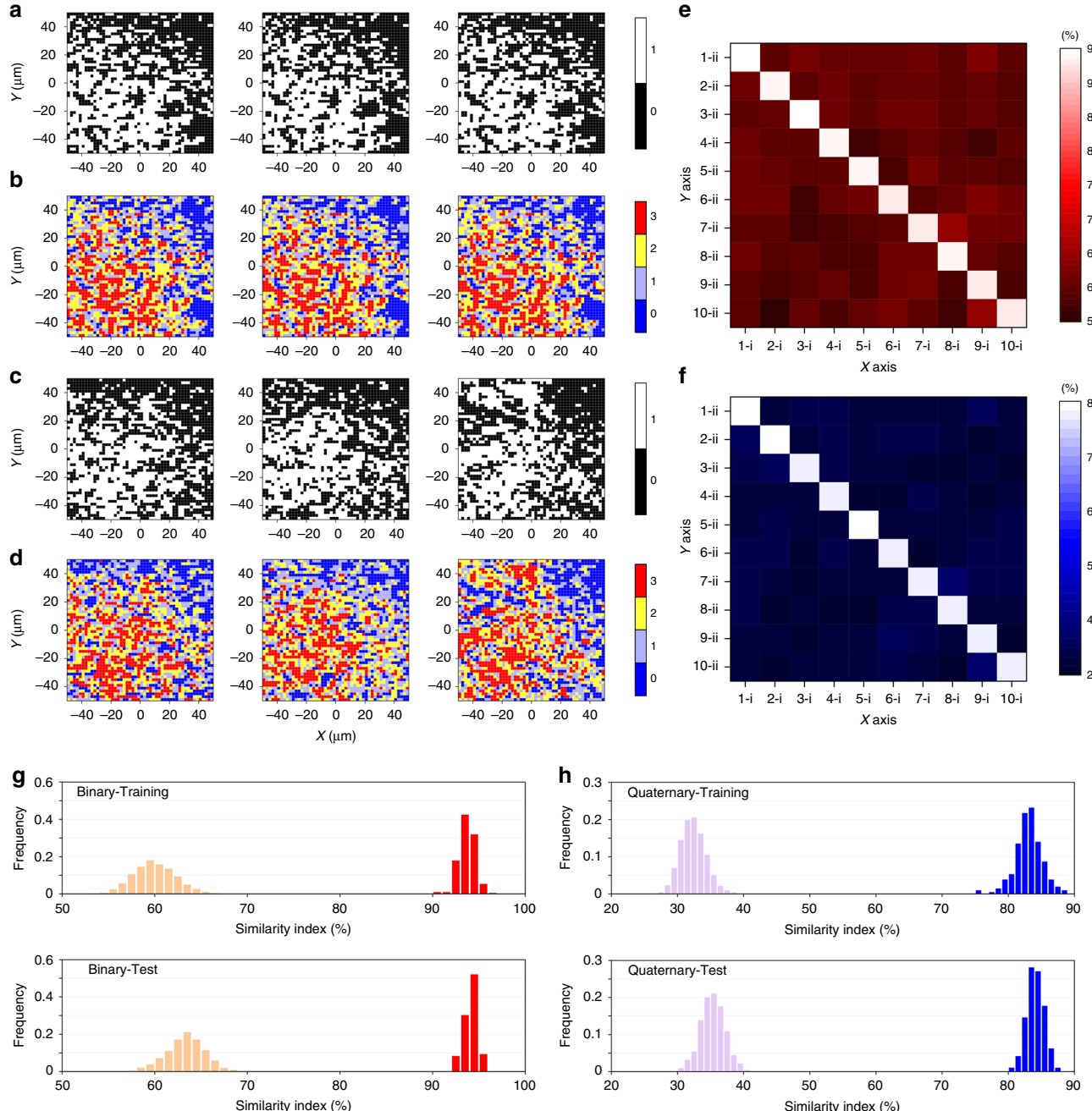

**Fig. 5 Reproducibility verification of PUF labels.** A PUF label composed of 4-NBT GERTs in a $100 \times 100 \ \mu m^2$ area is read with a resolution of $50 \times 50$ pixels and is digitized with **a** binary and **b** quaternary encoding of Raman intensity levels at each pixel for the first (left), second (middle), and third (right) measurement. Digitization of additional three labels numbered 2 (left), 3 (middle) and 4 (right) for **c** binary and **d** quaternary encoding of Raman intensity levels at each pixel with a resolution of $50 \times 50$ pixels. Pairwise match of ten PUF labels with **e** binary and **f** quaternary encoding of Raman intensity levels at each pixel. The x-axis and the y-axis represent the first and the second measurement of the labels, respectively, and the color bar shows the similarity index. Distribution of the similarity indexes ($I$) for the same PUF labels (red and blue bars) and those for different labels (orange and purple bars) of the training set (top) and the test set (bottom) in terms of **g** binary and **h** quaternary encoding of Raman intensity levels at each pixel. Figure **e** and **f** are plotted by referring to the ref. [17]. Source data are provided as a Source Data file.

mirrors, which rotate fast around two orthogonal axes, respectively. Third, the detector processes the collected data line by line in the SWIFT mode rather than pixel by pixel, thus greatly reducing transmission and processing time of data. It should be noted that the readout speed of Raman-based PUF labels at present still lags behind that of some other PUF labels, such as silicon PUFs[57], and is not enough yet to meet the requirement for the manufacturer's registration. Nevertheless, the

scanning speed of the lab-based confocal Raman system will be further continuously improved for practical use with many strategies and even the Raman mapping can be potentially realized on a hand-held device in the future. Apart from the synthesis of Raman tags with better performance, new Raman imaging modes could be introduced, such as the application of line-shaped[58,59] or multipoint laser[60–63], direct Raman imaging with a narrow-band filter[64], or a mode, where the stage

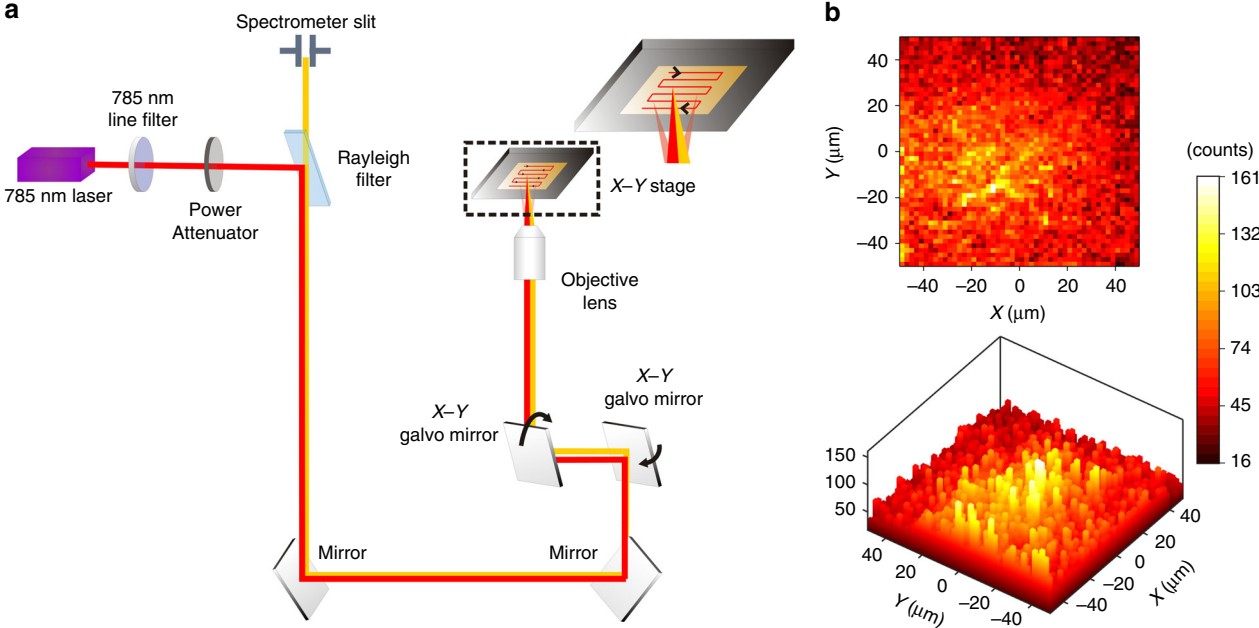

**Fig. 6 High-speed readout of PUF labels. a** Schematic of configurations of Raman system used for high-speed scanning. Under the DuoScan mode, two galvo mirrors rotate along two orthogonal axes in the mirror plane, controlling movements of the laser spot along the x- and the y-axis, respectively. **b** Two-dimensional (2D) (top) and 3D (bottom) plot for a PUF label ($100 \times 100 \, \mu m^2$) read in DuoScan mode in 6 s with a resolution of $50 \times 50$ pixels.

movement, light collection and data readout occur continuously and synchronously[58].

**PUF labels for practical applications**. To demonstrate the capability of practical use of GERTs-based PUF labels, we fabricated PUF labels on transparent Scotch tape (refer to Methods for more details), which can be transferred onto the surface of various products afterwards. The Scotch tape is selected also for minimizing damage to PUF labels from physical contact and environmental variation. After GERTs solution dried in the shape of a circular pattern on the adhesive side of the Scotch tape, we pasted the PUF label on a printing paper for demonstration (Fig. 7a). The PUF label ($1.8 \times 1.8 \, mm^2$) was read by Raman mapping (785 nm laser, $6 \times 10^4 \, W \, cm^{-2}$, 10× objective lens, exposure time of 10 ms per pixel) with a resolution of $50 \times 50$ pixels (Fig. 7b). It can be observed that the obtained 2D and 3D patterns from the mapping images coincide well with the circular pattern in the PUF label. We found that the background Raman signal from the Scotch tape and printing paper was negligible (e.g., point 1), whereas the signal from 4-NBT GERTs of the PUF label is strong (e.g., point 2). Therefore, we conclude that the Scotch tape is a suitable media to "pack" the Raman PUF labels for practical applications. The mapping images can be later digitized with the method mentioned previously. As the bare pattern constructed by NPs can be delicate to the environmental condition, we surely believe that the protection layer (e.g., polymer matrix) plays an important role in guaranteeing the physical robustness of PUF labels[17], which will be investigated in our future work. Figure 7c illustrates the application of the PUF label in the supply chain. PUF labels are first fabricated by manufacturers and then sold to goods producers for packaging. During the process, PUF labels are pasted onto commodities (e.g., drugs), read by a Raman spectrometer, digitized by software, and stored in database. During commodity circulation, the labels can be read, digitized, and authenticated at each stage, including distribution, retail and the end user. If a digital label does not match any in the database, it will be considered a fake. Since there may be differences in each reader in the supply chain, further investigation and optimization

should be implemented to determine reasonable error margins. Our extremely large encoding capacity offers significant protection against imperfections in PUF label fabrication and readout, to ensure the robustness of the method[8].

## Discussion

In this work, we have developed Raman-based PUF labels composed of different types of GERTs with various mapping resolutions for anticounterfeiting. In contrast to conventional fluorescent molecules and SERS tags[37], GERTs are more photostable for repeated authentication along the supply chain. Thanks to their 3D encoding capability (namely, the physical position of pixels, the type of GERTs, and the Raman intensity of selected bands), there is significant scope for further increasing the encoding capacity of GERTs PUF labels, for example, by increasing the mapping resolution to $100 \times 100$ pixels, though this would also increase the read time, leading to a tradeoff between resolution and scan duration. The improvement of SERS signals and the Raman system allows us to realize so-far the fastest scanning of GERTs-based PUF labels with an acquisition time down to 0.7 ms per pixel, and the reading speed could be further continuously increased in the future. In addition, our PUF system proves to be robust in distinguishing between different keys, which guarantees its security. Overall, this technique provides an effective way to combat against forgery and may be a viable solution for anticounterfeiting, especially as optical imaging technologies advance.

## Methods

**Materials and Instrumentation**. Chloroauric chloride (HAuCl$_4$•4H$_2$O) was acquired from Sinopharm Chemical Reagent Co. Ltd (Shanghai, China). Cetyl-trimethylammonium chloride (CTAC, 99%), 4,4'-biphenyldithiol (4,4'-BPDT, 98%), 2-mercapto-6-nitrobenzothiazole (2-M-6-NBT, 96%) and 2-naphthalenethiol (2-NT, 98%) were received from J&K Chemical Ltd (China). Ascorbic acid (>99%), 4-methylbenzenethiol (4-MBT, 98%), 2-chlorothiophenol (2-CBT, 98%) and 4-chlorothiophenol (4-CBT, 98%) were purchased from Aladdin (China). 2-Mercapto-5-nitrobenzimidazole (2-M-5-NBI, 97%) and 2-nitrobenzenethiol (2-NBT) were obtained from Sigma-Aldrich (Shanghai, China). 1,4-Benzenedithiol (1,4-BDT, 98%) and 4-nitrobenzenethiol (4-NBT) were acquired from TCI (Tokyo, Japan). All materials were used as received without any further purification.

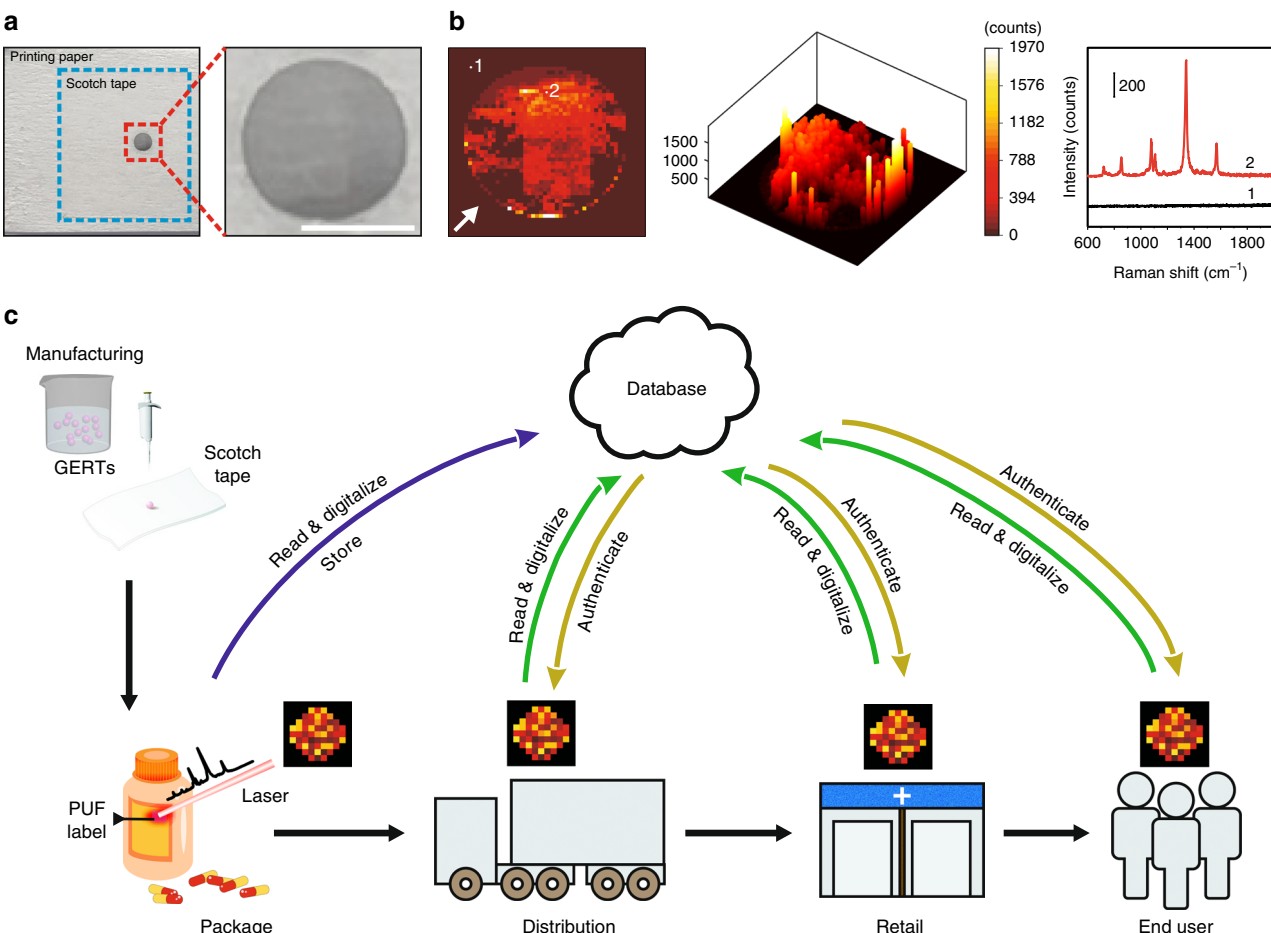

**Fig. 7 Demonstration of Raman PUF labels and their utilization in an anticounterfeiting system. a** A PUF label fabricated using 4-NBT GETRs on Scotch tape and **b** the corresponding 2D and 3D plots of the readout by high-resolution (50 × 50 pixels) Raman mapping. The background (Scotch tape and the printing paper) exhibits a negligible Raman signal (e.g., point 1) and the PUF label emits strong signals from its 4-NBT GERTs (e.g., point 2). Scale bar in panel **a** is 1 mm. **c** Schematic illustration of Raman PUF labels applied in the supply chain with an anticounterfeiting system. The composition of figure (**c**) refers to the ref. [16].

Ultrapure water (18.2 MΩ) was used for all experiments. Transmission electron microscope (TEM) images were collected on a JEM-2100F transmission electron microscope (JEOL, Tokyo, Japan) operated at 200 kV. Scanning electron microscope (SEM) images were obtained from a S-4800 scanning electron microscope (HITACHI, Tokyo, Japan). Ultraviolet (UV)–visible (Vis) extinction spectra were measured from a UV1900 UV–Vis spectrophotometer (Aucybest, Shanghai, China).

**Fabrication of PUF Tags**. 1,4-BDT embedded GERTs (1,4-BDT GERTs) and 4,4'-BPDT embedded GERTs (4,4'-BPDT GERTs) were synthesized according to our previous work[1]. For the synthesis of GERTs embedded with 2-NBT, 4-NBT, 2-M-5-NBI, and 2-M-6-NBT (2-NBT GERTs, 4-NBT GERTs, 2-M-5-NBI GERTs, and 2-M-6-NBT GERTs), Au cores (0.47 nM), synthesized in accordance with our previous protocol[47], were first washed by centrifugation to reduce the concentration of CTAC in solution from 100 to 20 mM to encourage adsorption of Raman reporters. Then 100 μL ethanol solution of Raman reporters (10 mM) was added dropwise to 2 mL Au cores under ultra-sonication. After a period of incubation (10 min for 4-NBT, 30 min for 2-M-5-NBI and 2-M-6-NBT, 1 h for 2-NBT), the obtained solution was centrifuged and washed three times and redispersed in 1 mL CTAC solution. Finally, 480 μL ascorbic acid (40 mM) and 960 μL reporter-modified Au cores (0.97 nM) were in turn added to a mixture of 16 mL CTAC (50 mM) and 800 μL HAuCl$_4$ (4.86 mM) rapidly under vigorous sonication. 4-MBT GERTs, 2-NT GERTs, 2-CBT GERTs and 4-CBT GERTs were synthesized by slightly modifying the above protocol. In all, 4 mM ethanol solution of Raman reporters (300 μL for 4-MBT, 2-CBT and 4-CBT, 200 μL for 2-NT) was incubated with 1 mL washed Au cores (1 nM) for 8 h, and then washed three times and redispersed in 0.5 mL CTAC. Afterwards, 20 mL CTAC (50 mM), 1 mL HAuCl$_4$ (4.86 mM), 1 mL ascorbic acid (40 mM), and 900 μL modified cores were mixed under violent sonication.

PUF labels were fabricated by drop-casting 2 μL of the above GERTs solution onto a silica substrate with metallic marks. The GERTs solution may include one,

three or ten types of PUF tags. All PUF labels were dried for Raman measurements. To demonstrate the practical applications of PUF labels composed of GERTs, we additionally fabricated PUF labels on a transparent Scotch tape, which can be easily transferred onto different surfaces. Two microliter solution of 4-NBT GERTs (10 nM) was drop-cast on the adhesive side of the Scotch tape and kept for drying in air at room temperature. Then the tape was pasted on a printing paper for demonstration. The whole pattern (1.8 × 1.8 mm²) was read under a 785 nm laser (6.1 × 10⁴ W cm⁻², 10× objective lens) with a resolution of 50 × 50 pixels and a 10-ms exposure time per pixel.

**Readout of PUF Labels**. PUF labels (100 × 100 μm) were read via a confocal Raman system (LabRAM XploRA INV, Horiba) with the response of Raman mapping signals. A 785 nm laser (3 × 10⁵ W cm⁻², 60× objective lens) was used with a 10- or 50-ms exposure time per pixel. There are two different imaging modes applied in the mapping, namely the STAGE mode and the DuoScan mode. The former collects Raman signals of each point of a sample by stage movement, while the latter is based on the movement of the laser spot between points. Under the STAGE mode, 8 s, 90 s, and 20 min were needed for Raman mapping with a resolution of 2 × 2, 10 × 10, and 50 × 50 pixels, respectively. Under the DuoScan (SWIFT) mode, the Raman mapping speed can be significantly increased by shortening the exposure time to around 0.7 ms and only 6 s was needed to accomplish the mapping with a resolution of 50 × 50 pixels.

**Digitization and authentication of PUF labels**. In this work, we used MATLAB R2017b to analyze the data including spectrum preprocessing, spectrum fitting and digitization. (i) Data preprocessing. Removing background and noise signals from the spectrum before analysis is essential as their effect can be significant. A Savitzky–Golay filter was firstly employed to smooth all spectra. The background signals were subtracted by fitting a fourth-degree polynomial with a threshold of 0.001. The baseline correction uses the asymmetric truncated quadratic as a cost

function to establish standards for noise removal. (ii) Spectral demultiplexing. The experimentally measured Raman spectrum at each pixel composed of multiple GERTs is well approximated by a linear sum of weighted pure components, thus the non-negative least squares (NNLS) method was employed to extract the SERS signal contribution of each component from the multiplexed signal. Additional polynomial terms are introduced to fit background signals that are not completely removed in the data preprocessing. Optimization of the fit requires calculating:

$$\min \frac{1}{2}\left( S - \sum_n w_n R_n - \sum_m a_m P_m \right)^2$$

$$s.t. \ w_n \geq 0$$

(1)

where $S$ is a measured spectrum, $R_n$ is a reference spectrum of component $n$ (pure GERTs), $w_n$ is the weight of $R_n$, $P_m$ is the $m$-order polynomial term, and $a_m$ is the weight of $P_m$. In NNLS, $w_n$ are prevented from being negative, as GERT particles only add, never subtract, signal. Low-order polynomial terms are used to account for small drifts of the background signal. In this work, $m$ varies from 0 to 2. Since the experiment was carried out on a silicon dioxide substrate, the silicon dioxide Raman spectrum was also included as a reference in the fitting process. (iii) Digitization: The PUF label is finally digitized by quantifying the Raman signals in each pixel. Quantification includes categorizing the GERT types and quantifying the intensity of the Raman signals. Owing to fluctuation of signals induced by environmental conditions or the Raman system (such as fluctuation of laser and optical misalignment), raw Raman signal matrices from each type of GERTs was standardized with the method of Z-score into sets with the mean of 0 and the standard deviation of 1. Next, the global search (GS) algorithm was employed to determine the common threshold for digitization and a 2-value or 4-value coarse-grained coding method was used to map each pixel value to a member of the set {0, 1} or {0, 1, 2, 3}, respectively. The optimization process of GS algorithm aims to find the most appropriate threshold for digitization, which should result in not only relatively high reproducibility for the same labels but also significant disparity between different ones, and the optimization function was set as:

$$\max \sum_{i=1}^{N} \sum_{j=i+1}^{N} \text{sign}(x_i, x_j) \times \text{Accu}(x_i, x_j)$$

$$\text{sign}(x_i, x_j) = \begin{cases} +1, & \text{if } x_i \text{ and } x_j \text{ are responses of the same label} \\ -1, & \text{if } x_i \text{ and } x_j \text{ are responses of the different labels} \end{cases}$$

(2)

$$\text{Accu}(x_i, x_j) = \frac{\text{The pixel number with the same value between } x_i \text{ and } x_j}{\text{pixel number}}$$

where $N$ is the number of matrices used in the optimization process. The digitized labels could be recorded as a matrix in the format of $\begin{bmatrix} a_{11} & \cdots & a_{1n} \\ \vdots & \ddots & \vdots \\ a_{m1} & \cdots & a_{mn} \end{bmatrix}$, where $m$ stands for the number of pixels, $n$ represents the type of GERTs, and the value of element corresponds to the Raman intensity level. (iv) Authentication: In this part, a total of 100 PUF labels consisting of 4-NBT GERTs with a resolution of $50 \times 50$ pixels were scanned. Each label has been measured three times. The 70 PUF labels were used to find the threshold with the help of global optimization algorithm described in (iii) and the other 30 PUF labels were used to verify that the threshold is reasonable. Herein, $N = 70 \times 3 = 210$. It took about 2 min to calculate the best common threshold with $-0.2059$ and $-0.7565/-0.2567/0.5998$ for binary and quaternary encoding, respectively. The digitized matrices then go through pairwise comparison to get similarity indexes $I$ of the same labels (300 sample capacity) and those of different ones (21,735 sample capacity) for comparison. The similarity index $I$ can be mathematically defined as follows:

$$\text{Label A} = \text{Matrix } \mathbf{A} = \begin{bmatrix} a_{11} & \cdots & a_{1n} \\ \vdots & \ddots & \vdots \\ a_{m1} & \cdots & a_{mn} \end{bmatrix} \text{Label B} = \text{Matrix } \mathbf{B} = \begin{bmatrix} b_{11} & \cdots & b_{1n} \\ \vdots & \ddots & \vdots \\ b_{m1} & \cdots & b_{mn} \end{bmatrix}$$

$$\text{Matrix } \mathbf{C} = \mathbf{A} - \mathbf{B} = \begin{bmatrix} a_{11} - b_{11} & \cdots & a_{1n} - b_{1n} \\ \vdots & \ddots & \vdots \\ a_{m1} - b_{m1} & \cdots & a_{mn} - b_{mn} \end{bmatrix}$$

$$I = \frac{\text{The pixel number of } '0' \text{ of Matrix } \mathbf{C}}{\text{The total pixel number of Matrix } \mathbf{C}}$$

(3)

## Data availability
Data are available within the article and supplementary files. The source data underlying Fig. 5g, h are provided as a Source Data file. All other data that support the findings of the study are available from the corresponding author upon reasonable request.

## Code availability
Computer code used for preprocessing the data and digitizing the data is available in MATLAB R2017b algorithm library.

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

## Acknowledgements

We acknowledge the financial support from the National Natural Science Foundation of China (No. 81871401, 81571763, and 81622026), Shanghai Jiao Tong University (No. YG2016MS51, YG2017MS54 and YG2019QNA28), the State Key Laboratory of Onco-genes and Related Genes (No. 91-17-28), Shanghai Key Laboratory of Gynecologic Oncology, Guangci Professorship Program of Ruijin Hospital, the Science and Technology Commission of Shanghai Municipality (No. 19441905300), and Innovation Research Plan supported by Shanghai Municipal Education Commission (No. ZXWF082101).

## Author contributions

Y.G. and J.Y. conceived the idea for this work and designed the experiments. Y.G. and C.H. performed the experiments. Y.Z. and L.L. assisted with sample preparation and the measurements. Y.G., C.H. and J.Y. analyzed the data. Y.G., C.H., B.T. and J.Y. wrote and revised the manuscript. All authors discussed the results and commented on the manuscript at all stages.

## Competing interests

The authors declare no competing interests.
