## [Peer Review File · Nature Communications]

Reviewers' comments:

Reviewer #1 (Remarks to the Author):

Dear Ye and co-workers

They say that duplication is the most sincerest form of flattery, so thank you.

I have gone over your work in detail and I am impressed by how you have managed to create a 11 response PUF key using surface enhanced Raman tags. You clearly show that it is fully feasible.

I would like you to go over all my comments in the attached file, and address the raised therein.

Prior to publication I would like you to try and find a method of analysis the is size robust and where you can define a common threshold for all PUF keys. If possible, an illustration like Figure 7 in

Arppe-Tabbara, R.; Tabbara, M.; Sørensen, T. J., Versatile and Validated Optical Authentication System Based on Physical Unclonable Functions. ACS Applied Materials & Interfaces 2019, 11 (6), 6475-6482.

would really help to showcase your work.

I look forward to seeing a revised version of the manuscript with a robust parameter replacing 'I'. That is the only part I feel is missing before I can recommend this manuscript for publication.

Best regards
Thomas

Reviewer #2 (Remarks to the Author):

In the received manuscript, the authors clearly reported their advances in inventing a platform to generate physically unclonable labels used for anticounterfeiting. It's convincing that gap-enhanced Raman tags (GERTs) with embedded Raman reporters could offer both high sensitivity and high photostability. More importantly, when barcoding with Raman reporters combines with drop-casting fabrication, the coding capacity gets remarkably enhanced in both spectral and special dimensions.

However, I believe the novelty of this study is not sufficient enough yet to get published on Nature Communications. The concept of Raman barcoding has been reported and well explored (DOI:10.1039/C4CS00382A, DOI:10.1039/C0CC04415F, DOI:10.1039/C4RA16163G and DOI: 10.1038/NMETH.4578) and some of these pioneer work showed even higher coding capacity in spectral domain. I think the key to a practical Raman-based PUF labels is not merely high coding capacity but robustness. The authors payed lots of efforts to prove the high capacity which I think it's not necessary. In conclusion, I suggest a Major revision to this paper. More specific comments are shown as following:

1. The authors demonstrated drop-casting with 1, 3 and 10 type of GERTs in the manuscripts. I think the demonstration of 3 GERTS are redundant as the ultimate purpose of this work is to produce label with 10 GERTS and this demonstration didn't offer more information and guidance for following work.

2. As the coding capacity can be theoretically proved to be extremely high, I believe the authors need to shift their experimental focus to demonstrate the robustness of readout. It appears to me

that the special pattern generated by this drop-casting platform is delicate especially to external force like scratch. Could the author show how these labels can be protected and preserved against harsh conditions such as light exposure, oxidation and force? These demonstrations could make this invention completer and more significant.

3. It's known the mapping speed of confocal Raman is not yet satisfactory for large area imaging, which could hinder the practical use of Raman barcode. As this is the key challenge of Raman PUF labels, the author ought to provide more details on this side in the manuscript instead of just mentioning it in SI. Our overall feeling of this work is that the authors' focus is on merely capacity, which has been well proved in other works. The key challenges like readout speed and robustness are somehow circumvented. This is the main concern that we believe the novelty of this paper is not sufficient.

Reviewer #3 (Remarks to the Author):

Summary: The authors introduced a novel way to construct an optical PUF using gap enhanced Raman tags. The proposed PUF can be used as a tag on products to secure the supply chain.

Strengths:

1. The authors fabricated the proposed PUFs.
2. The authors conducted a lot of experiments on the fabricated PUFs to show its large encoding capacity (response space in some PUF literature). To enlarge the encoding capacity of the proposed PUF, the authors applied many techniques: quaternary encoding and more types of SERS nanoparticles in the fabrication process.
3. To show its practicality, the authors also did an authentication experiment when the PUF is on printing paper.
4. The authors also improved the read-out speed of the PUF.

Weaknesses:

1. One of the major objectives of the authors is to enlarge the encoding capacity of the PUF. This is not the only concern of a PUF system. A PUF with larger encoding capacity can reduce the probability of fabricating two tags by a stochastic process and resulting in the same response. We also need to understand the distribution of the responses in the encoding capacity, otherwise, it may still be relatively easy for an attacker to fabricate another PUF that has the same response or has a close enough response which can lead to a false positive. The authors also mentioned that the distribution of nanoparticles is not even on the whole surface, then it will be very helpful to characterize the distributions of nanoparticles in every pixel. From the statistical study of every pixel, we will know how much randomness exists in each pixel and how much security we can get from the large encoding capacity. In other words, although the encoding capacity of the proposed PUF is very large, the probability of each response is not equal, so an attacker can use the response which has the highest probability of being fabricated to guess an arbitrary response. For example, in fig 3 f (iii), most of the pixels are encoded as 1, so the best guess for an attacker will be a response with all 1s.
2. In the authentication application, the authors need to clarify their attacker model. The authors need to specify what are the trusted components. For example, if the complete Raman system is not trusted at the verifier side, an attacker can use his own Raman system to measure the response of a legitimate tag and replay it to the verification server to pass the verification.

3. The authors improved the read-out speed of the PUF, such that it takes 6 seconds for measuring the response of a tag. Please justify the practicality of the proposed PUF when it is compared with a silicon PUF, which usually takes less than 1 milliseconds to measure, and does not require a complex optical system for measurement. Please give a reference to back up the claim that the total measurement time is 20 minutes.

4. The authors claimed that to use this tag in practice (improve the true positive rate and reduce the false positive rate), every tag needs its own optimum digitization threshold which will be stored in the verification server as well. Please elaborate on the process of finding this optimum threshold, and how long this process takes. Also, please comment on how this optimum threshold will affect the similarity index different PUFs and the same PUFs, and therefore effectively shrink the encoding capacity.

Editorial comments:

1. What do you mean by "It must be sufficiently large that the chance of generating two identical labels using the stochastic process is vanishingly small"?

2. On line 381: can be understand -> can be understood

3. Second Table S2 should be Table S3, and the caption of the second Table S2 is wrong.

4. It is highly recommended to insert lines in tables to separate every row.

Response to reviewers' comments

Reviewer 1

Comment 1: Prior to publication I would like you to try and find a method of analysis that is size robust and where you can define a common threshold for all PUF keys. If possible, an illustration like Figure 7 in Arppe-Tabbara, R.; Tabbara, M.; Sørensen, T. J., Versatile and Validated Optical Authentication System Based on Physical Unclonable Functions. ACS Applied Materials & Interfaces 2019, 11 (6), 6475-6482. would really help to showcase your work.

Response: Thanks so much for the great suggestions from the reviewer. We agree that the processing method of readouts should be improved to make our PUF system more robust. In the previous version, as indicated in below Figure R1a, the digitized labels have shown nonuniform distributions of intensity levels (many '0' in the matrices), which would significantly reduce the real encoding capacity of the PUF system. We carefully analyze the readouts and find out that this issue is most likely induced by the compression of many pixels into a small intensity range after normalization due to several ultra-bright pixels. Thus, we adopt a new method of **Z-score** to process the readout matrices. In statistics, the Z-score is the signed fractional number of standard deviations by which the value of an observation or data point is above the mean value of what is being observed or measured (see *E. Kreyszig (1979). Advanced Engineering Mathematics (Fourth ed.). Wiley. p. 880*). With this Z-score method, the matrix is standardized to a set with the mean of "0" and the standard deviation of "1". The Z-score method helps to significantly reduce the effects of the extremely small portion of the ultra-bright pixels in the whole matrix, resulting in much

more uniform distribution of different intensity levels. As a result, four intensity levels (0, 1, 2 and 3) are more uniformly distributed, as indicated in below Figure R1b and Figure 3f iii in the revision. Under this circumstance, the similarity index I , as a valid parameter, is applied. I is obtained from point-by-point comparison by calculating the percentage of the match pixels between two matrices. It can be defined mathematically as follows:

$$\text{Label A} = \text{Matrix A} = \begin{bmatrix} a_{11} & \cdots & a_{1n} \\ \vdots & \ddots & \vdots \\ a_{m1} & \cdots & a_{mn} \end{bmatrix}, \text{Label B} = \text{Matrix B} = \begin{bmatrix} b_{11} & \cdots & b_{1n} \\ \vdots & \ddots & \vdots \\ b_{m1} & \cdots & b_{mn} \end{bmatrix},$$

$$\text{Matrix C} = \text{A-B} = \begin{bmatrix} a_{11} - b_{11} & \cdots & a_{1n} - b_{1n} \\ \vdots & \ddots & \vdots \\ a_{m1} - b_{m1} & \cdots & a_{mn} - b_{mn} \end{bmatrix}$$

$$I = \frac{\text{The pixel number of '0' of Matrix C}}{\text{The total pixel number of Matrix C}}$$

Figure R1. Two-dimensional (2D) plots of digitization of the PUF label based on quaternary encoding of Raman intensity levels on each pixel with a resolution of 50×50 pixels using (a) the standard normalization method in the previous version and (b) a Z-score method.

Additionally, we have scanned much more PUF labels (totally 100 PUF labels) with a resolution of 50×50 pixels as the sample database for building up the robustness of our authentication system. Each label was read three times. Among the standardized readouts of 100 PUF labels, 70 labels are used to find common thresholds for digitization with the help of global optimization algorithm and the rest 30 ones were used to verify that the threshold is reasonable.

The optimization function is set as

$$\max \sum_{i=1}^N \sum_{j=i+1}^N \text{sign}(x_i, x_j) \times \text{Accu}(x_i, x_j)$$

$$\text{sign}(x_i, x_j) = \begin{cases} +1, & \text{if } x_i \text{ and } x_j \text{ are responses of the same label} \\ -1, & \text{if } x_i \text{ and } x_j \text{ are responses of the different labels} \end{cases}$$

$$Accu(x_i, x_j) = \frac{\text{The pixel number with the same value between } x_i \text{ and } x_j}{\text{pixel number}}$$

where N is the number of matrixes, here $N = 70 \times 3 = 210$. The obtained common thresholds here are -0.2059 and -0.7565/-0.2567/0.5998 for binary and quaternary coding, respectively. The digitized matrices then go through pairwise comparison to get similarity indexes I of the same labels (300 sample capacity) and those of different ones (21735 sample capacity). To find whether our PUF system can distinguish between different labels, we plot the distribution histograms of I from the 100 PUF labels (see below Figure R2 and Figure 5g, h in the revision), as suggested from the recommended paper by the reviewer. From the histogram, we can see similarity indexes of the same PUF labels can be well separated from those of different labels. Then, we preliminarily suggest an error margin of 85% for binary coding and 70% for quaternary coding according to the histograms, which could be further optimized with more sample data in real conditions. According to the article suggested by the reviewer (*ACS Appl Mater & Inter* 2019, 11, 6475), we have also plotted a similar illustration as shown in below Figure R3 and Figure 5e, f. This figure shows the validation of 10 randomly selected PUF labels between their first and second measurement to clearly demonstrate the robustness of the authentication algorithm. The sharp contrast of the matrices indicates great disparity between I of the same labels and that of different ones, which means our PUF system can successfully fabricate unique labels.

Therefore, we have greatly improved the robustness of our authentication system by applying the Z-core method and building up more suitable error margins/thresholds, and we have revised our manuscript accordingly as follows.

Figure 3 and Figure 5 in the revision have been revised or redrawn for the demonstration of robustness of the authentication system.

Page 13: “First, the raw Raman signals (intensity at 1078 cm^{-1}) was processed with a method of Z-score. In statistics, the Z-score is the signed fractional number of standard deviations by which the value of an observation or data point is above the mean value of what is being observed or measured [52]. With this Z-score method, the raw readout matrices were standardized to a set with the mean of “0” and the standard deviation (SD) of “1” for both binary and quaternary encoding.”

Page 14: “In real conditions, the encoding capacity of the PUF label would shrink if an error

margin is introduced for authentication, and the shrinkage has relations with pattern distribution. The digitized mappings in Fig. 3e and Fig. 3f show uniform distributions of different intensity levels, especially the ones with higher resolutions, which is in favor of a large real encoding capacity.”

Page 20: “To simulate the authentication process in real conditions, a total of 100 different PUF labels were selected for measurements. The readout of each label was repeated three times and the obtained matrices were standardized with the method of Z-score. Afterwards, a training set of 70 PUF labels was used for the search of common thresholds for digitization through the global optimization algorithm and the test set of the rest 30 labels was used to verify whether the thresholds are reasonable (see more details in Methods). Briefly, the most appropriate thresholds should result in not only relatively high reproducibility for the same labels but also significant disparity between different ones. The digitized matrices then went through pairwise comparison to get similarity indexes I of the same PUF labels and those of different ones. First, four labels (Fig. S10) are chosen as examples to demonstrate the digitizing effects, as presented in Fig. 5a-d (see Fig. S11 for 3D version), where all digital patterns show uniform distributions of various intensity levels. Herein, we use matrices with a resolution of 50×50 pixels for authentication because higher resolutions tend to give rise to fewer false positives [17]. It can be observed that the three digitized patterns from three measurements of the same label have a high degree of resemblance (Fig. 5a and 5b) and big similarity indexes ($I_{11^{\cdot}1}$, $I_{11^{\cdot}2}$ and $I_{11^{\cdot}3}$) of around 94% and 84% for binary and quaternary encoding, respectively (Table S2). The fact that the match does not reach 100% can most likely be explained by instability of the Raman system (including laser and optical alignment) and signal fluctuation from the SERS NPs.”

Page 21-22: “When it comes to the digitization of additional three labels numbered 2, 3 and 4 (Fig. 5c and 5d), obvious disparity could be observed between these labels and the one displayed in Fig. 5a and 5b. Their similarity indexes (I_{12} , I_{13} and I_{14}) have significantly dropped to around 57% and 30% for binary and quaternary encoding, respectively (Table S2). Then, we randomly selected 10 PUF labels for validation between their first and second measurement to clearly show the robustness of the authentication algorithm, as presented in Fig. 5e and 5f of a 10×10 matrix for binary and quaternary encoding, respectively. The sharp contrast of the matrices indicates great disparity between I of the same labels and that of different ones, which means that our PUF

system can successfully fabricate unique labels. To demonstrate the validation results more completely, the distribution histograms of similarity indexes are plotted (Fig. 5g and 5h) with a total sample capacity of 300 and 21735 for the same PUF labels and different ones. Both the training set and test set show that similarity indexes of the same labels are well separated from those of different labels with a gap of around 20% and 30% for binary and quaternary coding, respectively, indicating that it is possible to distinguish real labels from the duplicate ones. According to the histograms, here we preliminarily suggest an error margin of 85% and 70% for binary and quaternary encoding, respectively, which could be further optimized with more sample data in real conditions. In practical use, a best digitization threshold (or list of 3 thresholds for quaternary encoding) would be found by the manufacturer through the optimum algorithm with sufficient sample capacity of PUF labels. This threshold is then applied for the digitization of all PUF labels. At the point of authentication in the supply chain, the PUF label is scanned and digitized, and the label data is sent to be compared to labels in the data base. If a label is found in the database with a similarity index above the error margin, it is declared a match and the label is authenticated as genuine. Otherwise the label is rejected. The thresholds used for digitization and the error margin therefore play an important role in determining the security of the system and should be chosen carefully.”

Figure R2. Distribution of the similarity indexes (I) for the same PUF labels (orange and purple bars) and those for different labels (red and blue bars) with a resolution of 50×50 pixels of the training set (top) and the test set (bottom) in terms of (a) binary and (b) quaternary encoding of Raman intensity levels at each pixel.

Figure R3. Pairwise match of ten PUF labels with a resolution of 50×50 pixels in terms of (a) binary and (b) quaternary encoding of Raman intensity levels at each pixel. The X axis and the Y axis represent the first and the second measurement of the labels, respectively, and the color bar shows the similarity index.

Comment 2: I look forward to seeing a revised version of the manuscript with a robust parameter replacing 'I'. That is the only part I feel is missing before I can recommend this manuscript for publication.

Response: Thank the reviewer for the comments. Yes, we agree that *I* is not a good choice when the matrices have a large number of zeroes in the previous version. We carefully analyze the raw readouts and find out that the many zeroes are most likely due to the compression of the majority of pixels into a small range induced by several ultra-bright points during the standard normalization process. Thus, we have changed the processing method from normalization to Z-score standardization, and we found that our PUF system can differentiate between different labels very effectively. Therefore, we consider the similarity index of *I* is still a valid parameter in the revision of the manuscript. Please see more details from the **Response to Comment 1** of the **Reviewer 1**.

Below comments of A1-A38 are from the Reviewer 1 mentioned in the attached pdf file.

Comment [A1]: They are not used as but in labels. The tags are part of the label, not the label itself.

Response: Thanks for pointing out the inaccurate expression of the title. We really agree that the tags are just components of the label. **Therefore, we have changed the title into “Gap-enhanced Raman tags for physically unclonable anticounterfeiting labels”.**

Comment [A2]: I know acronyms are fashionable, but I strongly recommend that you do not invent a new acronym for existing technology. These tags are not new, so use something like Raman tag instead.

Response: Thanks for the suggestion. We agree with the reviewer very much. That is why in the introduction part and the beginning part of the Results, we only use the Raman tags instead of the acronym of GERTs. In fact, the acronym GERTs (gap-enhanced Raman tags) is not first defined in this work. It has actually been first-time defined for the core-shell nanostructure with the interior nanogap in our previous work (*ACS Appl. Mater. Inter.* 2017, 9, 3995), and has been discussed for a number of biomedical applications (see *Biomaterials* 2018, 163, 105; *Small* 2018, 14, 1801022; *ACS Nano* 2018, 12, 7974; *ACS Nano*, 2018, 12, 6492; *Nature Commun.*2019, *accepted*). Compared with traditional Raman tags, GERTs are much more appropriate for PUF labels mainly due to two factors: (1) a large enhancement factor down to a single-nanoparticle level (see *Nature Commun.*2019, *accepted*), which leads to strong readout signals with good signal-to-noise ratio, especially in high-speed imaging. (2) ultrahigh photostability (see *ACS Appl. Mater. Inter.* 2017, 9, 3995; *RSC Adv*, 2018, 8, 14434), which allows robust readouts during repeated authentications. **Therefore, we intend to use the term GERTs in the rest parts of the manuscript to emphasize the importance of GERTs for PUF labels.**

Comment [A3]: What is the matrix? Any polymer used to immobilize them? Some sort of physical resistance must be included in the system.

Response: Thanks very much for the suggestion. In fact, we are mainly focusing on the demonstrating the concept of GERTs-based PUF labels in this work. Thus, the label is fabricated by drop-casting 6 μ L solution containing GERTs on the silica substrate and letting the solution air-dry to form a circular pattern without any physical resistance or polymer matrix. When it comes to the demonstration for practical use, we use Scotch tape as the protective layer which proves to be an ideal coating as it produces negligible background Raman signals in our process.

We really agree with the reviewer that polymer immobilizations can give strong physical support to nanoparticles (NPs). We even consider that the PUF labels can be protected by a transparent plastic or glass cover in the practical applications. **We will investigate this effect in our**

future work. Therefore, we have added more discussion about the physical protection of GERTs-based PUF labels as follows.

Page 25: “As the bare pattern constructed by NPs can be delicate to the environmental condition, we surely believe that the protection layer (e.g., polymer matrix) plays an important role in guaranteeing the physical robustness of PUF labels [16], which will be investigated in our future work.”

Comment [A4]: This has to be mathematically proven, see Arppe Science Advance or our new paper: Arppe-Tabbara, R.; Tabbara, M.; Sørensen, T. J., Versatile and Validated Optical Authentication System Based on Physical Unclonable Functions. *ACS Applied Materials & Interfaces* **2019**, *11*(6), 6475-6482.

Response: Thanks for the advice and the recommendation of the excellent papers. The two relevant papers all give the effective encoding capacity of PUF labels which would be affected by particle size and distribution, readout noise and so on. Herein in our work, we just give the highest theoretical encoding capacity of GERTs-based PUF labels. For labels composed of ten types of GERTs with quaternary encoding of Raman intensity levels, each pixel has 4^{10} responses. When the mapping resolution is set as 50×50 pixels, we can get the highest theoretical encoding capacity of $(4^{10})^{2500}$ (namely, 3.2×10^{15051}). In practical use, when an error margin is introduced into the authentication system, the encoding capacity will be reduced accordingly.

Comment [A5]: Must be mentioned that this requires a lab based microscope.

Response: Thanks for pointing out the lack of clear statement of a lab-based Raman microscope required for (high-speed) Raman imaging.

Therefore, we have revised some sentences in the manuscript to emphasize the requirement of a lab-based Raman spectrometer for Raman mapping as follows.

Abstract: “A high-speed imaging mode on the confocal Raman system has been applied to significantly decrease the readout time of a PUF label with 50×50 pixels to 6 s. Authentication experiments have indicated the robustness and security of the PUF system.”

Page 6: “The PUF labels are read using a lab-scale confocal Raman system by performing Raman

mappings with different resolutions.”

Page 22: “For practical use, faster readout is usually required. To solve the problem, a high-speed ‘DuoScan’ mode (Fig. 6a) along with a ‘SWIFT’ mode on the confocal Raman system was used [55].”

Comment [A6]: Disclaim on how difficult it would be to realise a hand held device must be added.

Response: Thanks for pointing out the problem, and we have revised the sentence accordingly as follows.

Page 24: “Nevertheless, the scanning speed of the lab-based confocal Raman system will be further continuously improved for practical use with many strategies and even the Raman mapping can be potentially realized on a hand-held device in the future. Apart from the synthesis of Raman tags with better performance, new Raman imaging modes could be introduced, such as the application of line-shaped [57, 58] or multipoint laser [59-62], direct Raman imaging with a narrow-band filter [63], or a mode, where the stage movement, light collection and data readout occur continuously and synchronously [57]”.

Comment [A7]: You must add the references using surface enhanced Raman here. Take them from the Nat Rev Chem review.

Response: Thanks for pointing out the lack of references here. Therefore, we have added these references on page 3 (*ACS Appl. Mater. Interfaces*, 2016, 8, 9384; *Adv. Opt. Mater.*, 2016, 4, 1475; *J. Mater. Chem. C*, 2016, 4, 4312) and changed the discussion as follows.

Page 3: “Anticounterfeiting labels are one of the most common solutions, such as holograms, watermarks, graphical barcodes, and security inks that can only be inspected under ultraviolet light or by fluorescence microscopy. In addition, nanostructured-surface labels containing probes with surface-enhanced Raman scattering (SERS) signatures are modern extensions of conventional anticounterfeiting system, including Raman barcoding [*ACS Appl. Mater. Interfaces*, 2016, 8, 9384; *Adv. Opt. Mater.*, 2016, 4, 1475] and Raman patterning [*J. Mater. Chem. C*, 2016, 4, 4312]. More advanced authentication labels have been developed based on molecular tags, e.g., DNA, peptides, and polymers, which encode information mainly through the sequence of their

building blocks”.

Comment [A8]: Please cite Arppe-Tabbara, R.; Tabbara, M.; Sørensen, T. J., Versatile and Validated Optical Authentication System Based on Physical Unclonable Functions. *ACS Applied Materials & Interfaces* **2019**, *11*(6), 6475-6482. And Carro-Temboury, M. R.; Arppe, R.; Vosch, T.; Sørensen, T. J., An optical authentication system based on imaging of excitation-selected lanthanide luminescence. *Science Advances* **2018**, *4* (1), e1701384.

Response: Thanks for suggestion of the excellent papers. We have added these citations on page 4.

Comment [A9]: That is true theoretically, but the rate of false positives is the critical parameter for the application of a PUF system.

Response: Thank the reviewer for pointing out this issue. We agree that the rate of false positives is significant in the practical use of PUF labels when an error margin is introduced.

Therefore, we have revised the corresponding paragraph as follows.

Page 4: “The encoding capacity is fundamental...to be cracked [7]. It must be sufficiently large that the chance of generating two identical labels using the stochastic process is vanishingly small”
→ “The encoding capacity is fundamental...to be cracked [7]. Also, the rate of false positives plays a significant role in practical applications of PUF labels. The encoding capacity must be sufficiently large so that the chance of generating two identical labels using the stochastic process is vanishingly small”.

Comment [A10]: Please add appropriate references here.

Response: Thanks for pointing out the lack of references here. Accordingly, we have added the following references on page 5: *ACS Appl. Mater. Inter.* 2016, *8*, 4031–4041; *Adv. Mater.* 2016, *28*, 2330–2336. Both of them are relevant to Raman-based unclonable labels.

Comment [A11]: Our work with photostable lanthanides should be mentioned here.

Response: Thanks for suggestion of the relevant excellent paper. We agree that the comparison of the photostability for SERS and that for fluorescence here is not conveyed in an accurate manner.

Therefore, we have revised the corresponding sentence in the revision as follows.

Page 6: “Also, SERS tags tend to be more material-stable and photostable than conventional fluorescence molecules [37], though some lanthanide-based luminescent materials have been reported to show a quite high photostability [16].”

Comment [A12]: Please mention here that you are demonstrating a concept using a lab scale reader and there’s some way to go for a hand held device to be made/used.

Response: Thank the reviewer for the advice. Accordingly, we have revised the manuscript here. Please refer to the **Response to Comment [A5] of Reviewer 1** for more details.

Comment [A13]: What is the security of the read out? Do you always get the same result when you read a label? See: Arppe-Tabbara, R.; Tabbara, M.; Sørensen, T. J., Versatile and Validated Optical Authentication System Based on Physical Unclonable Functions. *ACS Appl. Mater. & Inter.* **2019**, *11*(6), 6475-6482.

Response: Thanks for pointing out the concern of the security of the readout. In the authentication part, we have read PUF labels three times to evaluate their reproducibility and found the readout is quite robust. Take one label as an example (see below Figure R4 and Fig. S10a). The mappings of three repeated measurements on the same label have a high degree of similarity, though they are not exactly the same. It is common that the readouts of the same label can be slightly different, as indicated in the recommended paper, most likely due to instability of the Raman system (including laser and optical alignment) and signal fluctuations from NPs. To further demonstrate the security our PUF system, we have validated a total of 100 labels and found that the system can distinguish between different labels quite robust (refer to the **Response to Comment 1 of Reviewer 1** for details). Therefore, we believe that the security issue of the system can be solved.

Figure R4. Readout of the PUF label presented in Figure 5a and b with a resolution of 50×50

pixels for the (i) first, (ii) second, and (iii) third measurement.

Comment [A14]: This is the critical parameter for the functionality of the PUF system.

Response: Thanks again. We agree very much with the reviewer on this point. **In the revised manuscript, we have changed the processing method of raw readouts from normalization to Z-score standardization to make I a much more robust comparison parameter for the effectiveness of the PUF system. Please see more details from the **Response to Comment 1** of the **Reviewer 1**.**

Comment [A15]: Our Nat Rev Chem paper should be cited as inspiration of this figure. It is extremely similar to a figure we reported there.

Response: Sorry for that. We do get inspirations from the Nat Rev Chem paper to plot this figure.

Therefore, we have added the citation to the caption of Figure 1.

Figure 1: “Fabrication and encoding of Raman PUF labels...Furthermore, the number of responses of each pixel is a power function of the number of Raman intensity levels and an exponential function of the type of SERS NPs. **This figure is plotted by referring to the ref. of [7].**”

Comment [A16]: This is clever thinking.

Response: Thanks very much for the comment.

Comment [A17]: But are they physically robust when you do not have a supporting polymer matrix?

Response: Thanks for the concern of the physical robustness of our PUF labels. We agree with the reviewer very much that the polymer matrix can improve the physical robustness of the SERS PUF labels. We will investigate these in future work and this work only focuses on the demonstration of the concept of the GERTs-based PUF labels.

Please refer to the **Response to Comment [A3]** of **Reviewer 1** for more details.

Comment [A18]: While fundamentally true the larger number of black spaces in the image e.g. figure 3g iii significantly reduces the capacity of the system as the most likely outcome is a ‘0’.

Response: Thanks again. We really agree with the reviewer that the large number of black spaces will reduce the capacity. This is most likely caused by several ultra-bright pixels in the whole matrix. Therefore, we have changed the processing method of raw readouts from normalization to the Z-score standardization to significantly reduce the effects of the extremely small portion of the ultra-bright pixels on the whole matrix. **As a result, much more uniform distributions of different intensity levels have been obtained. Please see more details from the Response to Comment 1 of the Reviewer 1.**

Comment [A19]: It should be mentioned we used this exact same approach with lanthanide luminescence previously. This work is a beautiful extension of our work using a different technique, but the concept is not new as presented here.

Response: Thanks for pointing out this issue. We agree with the reviewer that the approach of enlarging the encoding capacity of optical PUF labels by adding the dimension of NP types has been presented in the above excellent relevant paper.

Therefore, we have added the citation to the sentence as follows.

Page 16: “To further enlarge the encoding capacity, we added another dimension [16] for encoding: the type of SERS NPs. It is known that the number of responses per pixel increases exponentially with the number of types of GERTs used in label fabrication.”

Comment [A20]: But are they read in the same manner each time and can the system differentiate between different PUF keys?

Response: Thank you for the question. Yes, all labels with an area of $100 \times 100 \mu\text{m}^2$ are read in the same manner by performing Raman mappings with a power density of $3 \times 10^5 \text{ W/cm}^2$. To find whether our system can differentiate between different PUF keys, we have read a total of 100 labels for authentication, and the results turn out good with no false positives in validation. **Please refer to the Response to Comment 1 of Reviewer 1 for more details.**

Comment [A21]: Yes, thanks.

Response: Thank you very much for the comment.

Comment [A22]: Is that an established statistical parameter? If so please cite the describing work. If not, please describe how I is defined in mathematical detail.

Response: Thank the reviewer for pointing out the unclear definition of the similarity index I . Yes, this statistical parameter of similarity index has been used before (see *Nature*, 2010, 11, 12; *REMOTE SENS. ENVIRON.* 1998, 63, 95–100), as obtained from point-by-point comparison by calculating the percentage of the match pixel between two matrices. It can be defined mathematically as follows:

$$\text{Label A = Matrix A} = \begin{bmatrix} a_{11} & \cdots & a_{1n} \\ \vdots & \ddots & \vdots \\ a_{m1} & \cdots & a_{mn} \end{bmatrix}, \text{Label B = Matrix B} = \begin{bmatrix} b_{11} & \cdots & b_{1n} \\ \vdots & \ddots & \vdots \\ b_{m1} & \cdots & b_{mn} \end{bmatrix},$$

$$\text{Matrix C} = \text{A-B} = \begin{bmatrix} a_{11} - b_{11} & \cdots & a_{1n} - b_{1n} \\ \vdots & \ddots & \vdots \\ a_{m1} - b_{m1} & \cdots & a_{mn} - b_{mn} \end{bmatrix}$$

$$I = \frac{\text{The pixel number of '0' of Matrix C}}{\text{The total pixel number of Matrix C}}$$

Therefore, we have cited the corresponding references and also describe the mathematical definition in the Methods on page 30-31 in the revision.

Comment [A23]: So you are getting 10-50 % wrong according to table S2, and naturally the large matrices will be mostly zeroes which makes I a bad parameter for the images with many pixels. You will need to make a better comparison parameter that is size robust, and then suggest a threshold for when you parameter can be used to indicate a match and when it is a non-match, see Arppe-Tabbara, R.; Tabbara, M.; Sørensen, T. J., Versatile and Validated Optical Authentication System Based on Physical Unclonable Functions. *ACS Applied Materials & Interfaces* **2019**, 11(6), 6475-6482.

Response: Thanks very much for the concern and suggestion of the comparison parameter. We have almost solved this issue by using the Z-score method during the digitalization process. Afterwards we counted all the similarity indexes from the 100 PUF labels and plot the distribution histograms, we preliminarily suggest an error margin (herein we do not use “threshold” to avoid confusion with the “threshold” for digitization) of 85% for binary coding and 70% for quaternary coding, which can be further optimized with more sample data in real conditions. **Please see more**

details from the **Response to Comment 1 of Reviewer 1**.

Comment [A24]: This is due to the many instances ‘0’ in the large matrices.

Response: We really agree that the many instances “0” will contribute a lot to the small standard deviation (SD) of the similarity indexes of the same PUF labels. **Again, please refer to the Response to Comment 1 of Reviewer 1 for details.** We have altered the method of data standardization to get much more uniform distributions of the matrices. We also calculate new SDs of the similarity indexes of the same labels (50×50 pixels) with a sample capacity of 300, and get SDs of 0.88% and 1.88% for binary and quaternary encoding, respectively. Compared with the previous version where SDs are 1.64% and 0.91% when the resolution is 50×50 pixels, the new SDs are still small but exclude the significant effect of the many “0” in the matrices.

Comment [A25]: You have to analyze the information content of the PUF keys. If it is just a couple of % this is still a huge SD.

Response: Thanks for the suggestion. We agree with the review very much that the pattern content of PUF keys should be analyzed when discussing SD of similarity index. The SDs of 4.56% and 4.39% belong to PUF labels with a resolution of 10×10 pixels, so they are relatively large. In the previous version, there are many zeroes in the digitized matrices, which can contribute a lot to small SDs. Therefore, we have changed the processing method of the data to obtain much more uniform distributions of the matrices, and the acquired SDs of are still small. **Please see more details from the Response to Comment [A24] of Reviewer 1.**

Comment [A26]: This is good

Response: Thank you very much for the comment.

Comment [A27]: What is the threshold?

Response: Thanks for the question. We are sorry for lack of the statement of the threshold for digitization. In the authentication part of the new version, raw readouts of PUF labels are standardized by the Z-score method and then used to search the threshold via global optimization algorithm. **Please see more details from the Response to Comment 1 of Reviewer 1. The**

obtained thresholds for binary and quaternary encoding are -0.2059 and -0.7565/-0.2567/0.5998, respectively. We have added it into the Methods on page 30-31 in the revision.

Comment [A28]: Their own or per type?

Response: Thank you for the question. In the previous version, each PUF label has a unique threshold for digitization. However, we cannot agree with the reviewer more that it is not reasonable. Therefore, we have used the same thresholds to digitize all PUF labels (-0.2059 for binary coding and -0.7565/-0.2567/0.5998 for quaternary coding) in the new version. The common thresholds can digitize the labels into uniformly distributed patterns and distinguish between different PUF labels. **Please see more details from the Response to Comment 1 of Reviewer 1.**

Comment [A29]: This is not acceptable, there should be a better solution to this issue. This can be found in a stronger mathematical treatment of the read out, see for inspiration the ESI of: Carro-Temboury, M. R.; Arppe, R.; Vosch, T.; Sørensen, T. J., An optical authentication system based on imaging of excitation-selected lanthanide luminescence. *Science Advances* **2018**, *4* (1), e1701384.

Response: Thank you for pointing out the issue of the threshold and recommending the excellent paper. After carefully reading the paper and considering the features of our raw readouts, we adopt a new method of Z-score to standardize the readout matrices and digitize all the same type (with the same resolution and type of NPs) of PUF labels with common thresholds. **Please refer to the Response to Comment 1 of Reviewer 1 for details. Therefore, we believe that we have found out a good solution.**

Comment [A30]: A lowest common threshold should be given. This threshold should be size independent, see Arppe-Tabbara, R.; Tabbara, M.; Sørensen, T. J., Versatile and Validated Optical Authentication System Based on Physical Unclonable Functions. *ACS Applied Materials & Interfaces* **2019**, *11* (6), 6475-6482.

Response: Thanks for the suggestion of a common threshold (or error margin) and the relevant paper. We have measured 100 single-type GERTs based PUF labels by performing Raman

mappings three times each with a resolution of 50×50 pixels, and then standardize them to Z-score and digitize them with the same thresholds. After that, similarity indexes are required by pairwise comparison of the matrices. Inspired by the recommended paper, we have plotted the distribution histograms of the similarity indexes, which shows good separation of I for the same PUF labels and those for different ones (see the above Figure R2). **Herein, we suggest lowest common error margins of 85% for binary coding and 70% for quaternary coding, which could be further optimized with more sample data in real conditions. Please see more details from the Response to Comment 1 of Reviewer 1.**

Comment [A31]: Please comment on the fact that the manufacturer will need to register PUF labels with a speed of 100s per minut.

Response: Thanks again. Under the DuoScan mode along with SWIFT mode, a PUF label with a resolution of 50×50 pixels can be read within 6 s with acquisition time of 0.7 ms per pixel, which means that 10 PUF labels can be registered per minute. 0.7 ms is so far the shortest acquisition time of commercial electron-multiplying charge-coupled device to the best of our knowledge. However, we have to admit that this speed is not enough yet to meet the requirement for the manufacturer's registration of PUF labels. But we also believe that the scanning speed will be further continuously improved with the development of Raman system by synthesizing the Raman tags with better performance and inventing new Raman scanning mode.

Therefore, we have added the corresponding discussion as follows.

Page 24: "It should be noted that the readout speed of Raman-based PUF labels at present still lags behind that of some other PUF labels, such as silicon PUFs [56], and is not enough yet to meet the requirement for the manufacturer's registration. Nevertheless, the scanning speed of the lab-based confocal Raman system will be further continuously improved for practical use with many strategies and even the Raman mapping can be potentially realized on a hand-held device in the future. Apart from the synthesis of Raman tags with better performance, new Raman imaging modes could be introduced, such as the application of line-shaped [57, 58] or multipoint laser [59-62], direct Raman imaging with a narrow-band filter [63], or a mode, where the stage movement, light collection and data readout occur continuously and synchronously [57]."

Comment [A32]: Nice and simple solution for a demonstration.

Response: Thanks very much for the comment.

Comment [A33]: This is only true if a suitable match parameter and common threshold can be defined.

Response: Thanks again for the suggestion. **Please refer to the Response to Comment 1 of Reviewer 1 for details.**

Comment [A34]: Hand held device is unrealistic, please remove.

Response: Thanks for pointing out this issue. We agree that hand-held Raman devices at present cannot realize imaging. **Therefore, we have removed the diagram of the hand-held device in Figure 7c.**

Comment [A35]: Area c is not mentioned. When (c) is included a citation to our Science Adv paper should be given as it is clearly a strong source of inspiration for this cartoon.

Response: We are really sorry for the lack of citation to the Science Adv paper in Figure 7c. The paper does give us inspirations to plot Figure 7c.

Therefore, we have added the citation in the caption of Figure 7.

Figure 7 caption: “Demonstration of Raman PUF labels and their...(c) Schematic illustration of Raman PUF labels applied in the supply chain with an anticounterfeiting system. **Figure (c) is plotted by referring to the ref. of [25].”**

Comment [A36]: Is this the conclusion?

Response: Thanks for the question. No, this part is the Discussion according to the format of *Nature Communications*.

Comment [A37]: But compared to lanthanide luminescence you lose an order of magnitude in brightness and has lower photostability. Please comment and cite: Carro-Temboury, M. R.; Arppe, R.; Vosch, T.; Sørensen, T. J., An optical authentication system based on imaging of excitation-selected lanthanide luminescence. *Science Advances* **2018**, *4* (1), e1701384. For

stability etc consult: Arppe, R.; Carro-Temboury, M. R.; Hempel, C.; Vosch, T.; Just Sørensen, T., Investigating dye performance and crosstalk in fluorescence enabled bioimaging using a model system. *Plos One* **2017**, *12* (11), e0188359.

Response: Thanks for the concern of the photostability of our GERTs. According to our previous work, the GERTs have shown ultrahigh photostability (see *ACS Appl. Mater. Inter.* 2017, 9, 3995) and below Figure R5a. The figure shows that the Raman intensity of BDT GERTs with silica coating only declines by less than 20% upon a continuous laser irradiation with a power density of 4.71×10^5 W/cm² for 30 min. This is much more photostable than the fluorescent dyes used in the paper *Plos One* 2017, 12 (11), e0188359, where the dyes bleach very fast losing almost all the intensity within 100 s (see below Figure R5b). The luminescent materials used in the paper *Science Advance* 2018, 4 (1), e1701384 demonstrate much higher photostability than fluorescent dyes, which lose negligible intensity during 30 min laser irradiation (see below Figure R5c). However, the laser power densities for excitation of the luminescent materials are 8.6×10^2 W/cm² and 9.9×10^2 W/cm² for Eu@LTA and Tb@LTA, respectively, almost three orders of magnitude lower than for GERTs. Therefore, it's hard to directly compare the photostability of luminescent materials and GERTs. It is worth noting that the exposure time for one pixel of the PUF label is 10 ms. If the label goes through 100 repeated authentications, the total time of laser irradiation on one pixel is only 1 s, during which the intensity loss of GERTs can be ignored. **Thus, we believe that the photostability of GERTs is suitable for PUF labels.**

Figure R5. (a) Photostability of GERTs with mesoporous silica coating under continuous laser irradiation with different power densities for 30 min. Copy from *ACS Appl. Mater. Inter.* 2017, 9, 3995. (b) Photostability of Eu@LTA and Tb@LTA compared to fluorescein. Spectra were recorded with the CCD camera and a 40X air objective (40X 0.65 UIS2 Microscope Infinity PlanC N Plan Objective, Olympus). A total of 180 spectra with an exposure time of 10 s were acquired consecutively in a 30 min span. Eu@LTA, Tb@LTA and fluorescein were excited at 465 nm, 488 nm and 488 nm using laserpowers of 5.1 μ W, 6.5 μ W and 0.9 μ W, respectively. Copy from *Plos One* 2017, 12 (11), e0188359. (c) Bleaching of F18, MitoTracker Red and ATTO647N. The spectra were recorded in 1 s intervals for 100 s. The excitation laser powers for F18, MitoTracker Red and ATTO647N are 7.2 μ W, 1.2 μ W and 2.9 μ W, respectively. Copy from *Science Advance* 2018, 4 (1), e1701384.

Comment [A38]: The concept is good, but the size, physical read time, method of analysis and the hardware demands are a problem for real world application. This is a scientific paper, here only the lack of a robust analysis is a problem.

Response: Thank you very much for pointing out the concerns of our PUF system. To improve the robustness of PUF labels, in the revision we have changed the processing method of raw readouts, digitized the labels with common thresholds, and preliminarily suggested error margins to indicate a match or non-match. And the authentication results show that our PUF system can distinguish between different labels very well. Please see details from the **Response to Comment 1 of Reviewer 1.**

We really agree with the reviewer that there is still a long way to go before GERTs-based PUF labels can be applied in real world. We believe that these labels will become more suitable for practical applications with the improvement of the label itself, the analysis method of data, and the Raman system.

Reviewer 2

Comment 1: However, I believe the novelty of this study is not sufficient enough yet to get published on Nature Communications. The concept of Raman barcoding has been reported and well explored (DOI:10.1039/C4CS00382A, DOI:10.1039/C0CC04415F, DOI:10.1039/C4RA16163G and DOI: 10.1038/NMETH.4578) and some of these pioneer work showed even higher coding capacity in spectral domain. I think the key to a practical Raman-based PUF labels is not merely high coding capacity but robustness. The authors payed lots of efforts to prove the high capacity which I think it's not necessary.

Response: Thank the reviewer for pointing out the concern about the novelty of our work. According to the recommendation of the reviewer, we have carefully read the following articles: (1) *Chem. Soc. Rev.*, 2015, 44, 5552. (DOI:10.1039/C4CS00382A); (2) *Chem. Commun.*, 2011, 47, 2306. (DOI:10.1039/C0CC04415F); (3) *RSC Adv.*, 2015, 5, 13762. (DOI:10.1039/C4RA16163G); (4) *Nat. Method*, 2018, 15, 194. (DOI: 10.1038/NMETH.4578). The first three articles mainly focused on the preparation of various SERS nanoprobe for multiplexing detection based on the Raman spectral fingerprints. These SERS nanoprobe were indeed used for barcodes but not for PUF labels, which are typically fabricated using stochastic processes to form random patterns constructed by disordered distributions of micro- or even nanostructures. The last article focused on the synthesis of various Raman molecular probe for stimulate Raman spectroscopy (SRS). These molecular probe were again not for PUF labels. In addition, SRS typically has much lower enhancement than SERS and consequently it is more challenge to realize the single-nanoparticle detection, which is important in the SERS nanoprobe-based PUF labels. We consider one of the innovative points in this work is first time (to the best of our knowledge) to achieve super-large the encoding capacity based on the three-dimensional encoding methodology with the combination of the type of SERS nanoprobe, the physical position of each pixel, and the level of

Raman intensities. We are not focusing on the enlarging the encoding capacity of SERS nanoprobe in Raman spectral domain in this work, although we have the capability to realizing more than 10 types of GETRs.

We are sorry to have given the reviewers a false impression in the previous version due to the unclear description that we only emphasize the importance of encoding capacity. Herein we summarize the novelties in this work as follows. (1) **Super-large encoding capacity**. To the best of our knowledge, we are first-time to achieve an encoding capacity of over 3×10^{15051} based on the three-dimensional encoding methodology with the combination of the type of SERS nanoprobe, the physical position of each pixel, and the level of Raman intensities. (2) **Fabricating the most suitable SERS nanoprobe, namely, GERTs for PUF labels**. According to our previous work, the GERTs are favorable for PUF labels because of not only a large enhancement factor down to a single-nanoparticle level (*see Nature Commun.2019, accepted*) but also ultrahigh photostability for repeated scanning (see the above Figure R5a and *ACS Appl. Mater. Inter. 2017, 9, 3995*). GERTs exhibit strong Raman enhancement with the combination of electromagnetic enhancement and chemical enhancement mechanism at the off-resonance condition, which greatly minimize the photothermal effect and the corresponding photobleaching effect (*ACS Appl. Mater. Inter. 2017, 9, 3995*). Moreover, the Raman reporter molecules are embedded in the core-shell nanogaps and isolated from the oxygen and moisture by the metallic shell. All these advantageous features we discovered before offer the ultrahigh photostability of GERTs. However, **SERS hot spots from the typical dimers or aggregates are apt to photobleaching** (see below Figure R6; *Science, 2008, 321, 388*; *ACS Appl. Mater. Inter. 2017, 9, 3995*), **indicating that they are inappropriate for the PUF labels**. (3) **Feasibility of GERTs-based PUF labels**. We realized so-far the fastest Raman mapping in a DuoScan mode (Xplora INV, Horiba), where the laser is scanned across the label in *X* and *Y* directions by the rotation of galvo mirrors instead of moving the stage mechanically. The Raman mapping with a resolution of 50×50 pixels can be acquired within an ultrashort acquisition time of 0.7 ms per pixel and 6 s in total. 0.7 ms is so far the shortest acquisition time of commercial electron-multiplying charge-coupled device (EMCCD) to the best of our knowledge. Such GERTs allows to largely improve the imaging speed **one to two orders of magnitude faster** compared to the previous work performed with a similar image resolution and laser power (*Nano Lett. 2015, 15,*

1766). Such high-speed readout is very close to the requirement in the practical authentication system. In addition, to demonstrate the capability of practical use of GERTs-based PUF labels, we fabricated PUF labels on transparent Scotch tape, which can be transferred onto the surface of various products afterwards. The Scotch tape is selected also for minimizing damage to PUF labels from physical contact and environmental variation. All these optimizations were performed to prove the feasibility of GERTs-based PUF labels. (4) **Robustness demonstration of PUF labels.** In the new version, we have applied a new method of **Z-score** to process the readout matrices. The digitized labels have shown uniform distributions of intensity levels (see Figure R1). In addition, we have scanned much more PUF labels (totally 100 PUF labels) with a resolution of 50×50 pixels as the sample database for building up the robustness of our authentication system. Each label was read three times. The obtained common thresholds here are -0.2059 and -0.7565/-0.2567/0.5998 for binary and quaternary coding, respectively. The digitized matrices then go through pairwise comparison to get similarity indexes I of the same labels (300 sample capacity) and those of different ones (21735 sample capacity). The results show that similarity indexes of the same PUF labels can be well separated from those of different labels, as indicated by the distribution histograms of I from the 100 PUF labels (see Figure R2 and Figure 5e, f in the revision).

Please refer to the **Response to Comment 1 of Reviewer 1** for more details.

Figure R6. (Left) Schematic diagrams, (middle) photostability measurements of time-resolved SERS spectra of 1,4-BDT coated gold nanoparticle aggregates on a silicon wafer during continuous irradiation for 30 min, and (right) three representative SERS spectra at selected irradiation times. SERS spectra were acquired at 2-s intervals (10-ms exposure time/spectrum and 100 \times objective lens). Adapted from *ACS Appl. Mater. Inter.* 2017, 9, 3995.

Accordingly, we have modified the manuscript to emphasize the all aforementioned novelties in

the revision as follows.

Figure 3 and Figure 5 in the revision have been revised or redrawn for the demonstration of robustness of the authentication system.

Page 10: “Consequently, their SERS signals are easily affected by the NP states and the environmental conditions (e.g., oxygen, moisture).”

Page 13: “First, the raw Raman signals (intensity at 1078 cm^{-1}) was processed with a method of Z-score. In statistics, the Z-score is the signed fractional number of standard deviations by which the value of an observation or data point is above the mean value of what is being observed or measured [52]. With this Z-score method, the raw readout matrices were standardized to a set with the mean of “0” and the standard deviation (SD) of “1” for both binary and quaternary encoding.”

Page 14: “In real conditions, the encoding capacity of the PUF label would shrink if an error margin is introduced for authentication, and the shrinkage has relations with pattern distribution. The digitized mappings in Fig. 3e and Fig. 3f show uniform distributions of different intensity levels, especially the ones with higher resolutions, which is in favor of a large real encoding capacity.”

Page 20: “To simulate the authentication process in real conditions, a total of 100 different PUF labels were selected for measurements. The readout of each label was repeated three times and the obtained matrices were standardized with the method of Z-score. Afterwards, a training set of 70 PUF labels was used for the search of common thresholds for digitization through the global optimization algorithm and the test set of the rest 30 labels was used to verify whether the thresholds are reasonable (see more details in Methods). Briefly, the most appropriate thresholds should result in not only relatively high reproducibility for the same labels but also significant disparity between different ones. The digitized matrices then went through pairwise comparison to get similarity indexes I of the same PUF labels and those of different ones. First, four labels (Fig. S10) are chosen as examples to demonstrate the digitizing effects, as presented in Fig. 5a-d (see Fig. S11 for 3D version), where all digital patterns show uniform distributions of various intensity levels. Herein, we use matrices with a resolution of 50×50 pixels for authentication because higher resolutions tend to give rise to fewer false positives [17]. It can be observed that the three digitized patterns from three measurements of the same label have a high degree of resemblance (Fig. 5a and 5b) and big similarity indexes ($I_{11^{\cdot}1}$, $I_{11^{\cdot}2}$ and $I_{11^{\cdot}3}$) of around 94% and 84% for

binary and quaternary encoding, respectively (Table S2). The fact that the match does not reach 100% can most likely be explained by instability of the Raman system (including laser and optical alignment) and signal fluctuation from the SERS NPs.”

Page 21-22: “When it comes to the digitization of additional three labels numbered 2, 3 and 4 (Fig. 5c and 5d), obvious disparity could be observed between these labels and the one displayed in Fig. 5a and 5b. Their similarity indexes (I_{12} , I_{13} and I_{14}) have significantly dropped to around 57% and 30% for binary and quaternary encoding, respectively (Table S2). Then, we randomly selected 10 PUF labels for validation between their first and second measurement to clearly show the robustness of the authentication algorithm, as presented in Fig. 5e and 5f of a 10×10 matrix for binary and quaternary encoding, respectively. The sharp contrast of the matrices indicates great disparity between I of the same labels and that of different ones, which means that our PUF system can successfully fabricate unique labels. To demonstrate the validation results more completely, the distribution histograms of similarity indexes are plotted (Fig. 5g and 5h) with a total sample capacity of 300 and 21735 for the same PUF labels and different ones. Both the training set and test set show that similarity indexes of the same labels are well separated from those of different labels with a gap of around 20% and 30% for binary and quaternary coding, respectively, indicating that it is possible to distinguish real labels from the duplicate ones. According to the histograms, here we preliminarily suggest an error margin of 85% and 70% for binary and quaternary encoding, respectively, which could be further optimized with more sample data in real conditions. In practical use, a best digitization threshold (or list of 3 thresholds for quaternary encoding) would be found by the manufacturer through the optimum algorithm with sufficient sample capacity of PUF labels. This threshold is then applied for the digitization of all PUF labels. At the point of authentication in the supply chain, the PUF label is scanned and digitized, and the label data is sent to be compared to labels in the data base. If a label is found in the database with a similarity index above the error margin, it is declared a match and the label is authenticated as genuine. Otherwise the label is rejected. The thresholds used for digitization and the error margin therefore play an important role in determining the security of the system and should be chosen carefully.”

Comment 2: The authors demonstrated drop-casting with 1, 3 and 10 type of GERTs in the

manuscripts. I think the demonstration of 3 GERTS are redundant as the ultimate purpose of this work is to produce label with 10 GERTS and this demonstration didn't offer more information and guidance for following work.

Response: We entirely agree with the reviewer. We have moved the relevant figures and descriptions of 3 types of GERTs into the Supplementary Information and have more emphasized the importance of 10 types of GERTs for PUF demonstration in the revision as follows.

Page 16: "To further enlarge the encoding capacity, we added another dimension [16] for encoding: the type of SERS NPs. It is known that the number of responses per pixel increases exponentially with the number of types of GERTs used in label fabrication. ... PUF labels with an area of $100 \times 100 \mu\text{m}^2$ consisting of three (Fig. S5 and S6) or ten (Fig. 4 and Fig. S7-9) types of GERTs were fabricated to verify its feasibility. Herein, a ten-type GERT PUF label is demonstrated as an example, which can be created by following a similar procedure described above."

Comment 3: As the coding capacity can be theoretically proved to be extremely high, I believe the authors need to shift their experimental focus to demonstrate the robustness of readout. It appears to me that the special pattern generated by this drop-casting platform is delicate especially to external force like scratch. Could the author show how these labels can be protected and preserved against harsh conditions such as light exposure, oxidation and force? These demonstrations could make this invention complete and more significant.

Response: Thank the reviewer for raising the concern about the robustness of the PUF labels. We indeed consider the physical robustness of the PUF labels and therefore we use the Scotch tape as a protection layer against harsh conditions.

We agree with the reviewer very much that the robust readout is significant for PUF system. We even consider that the PUF labels can be protected by a transparent plastic or glass cover in the practical applications. We will investigate these in future work and this work only focuses on the demonstration of the concept of the GERTs-based PUF labels.

Please refer to the **Response to Comment [A3] of Reviewer 1** for more details.

Comment 4: It's known the mapping speed of confocal Raman is not yet satisfactory for large

area imaging, which could hinder the practical use of Raman barcode. As this is the key challenge of Raman PUF labels, the author ought to provide more details on this side in the manuscript instead of just mentioning it in SI. Our overall feeling of this work is that the authors' focus is on merely capacity, which has been well proved in other works. The key challenges like readout speed and robustness are somehow circumvented. This is the main concern that we believe the novelty of this paper is not sufficient.

Response: Thank for the reviewer's suggestion. We cannot agree with the reviewer more and we have moved the scheme of high-speed imaging mode and the description of Raman mapping method from the SI into the main text of the manuscript (see below Figure R7 and Figure 6 in the revision). In addition, we have added more relevant discussion about the high-speed Raman mapping in the revision as follows.

Figure 6 has been added to emphasize the details of high-speed Raman mapping method.

Page 23-24: "For practical use, faster readout is usually required. To solve the problem, a high-speed 'DuoScan' mode (Fig. 6a) along with a 'SWIFT' mode on the confocal Raman system was used [55]. To demonstrate high-speed readout in DuoScan mode, a PUF label ($100 \times 100 \mu\text{m}^2$) fabricated using 4-NBT GERTs was mapped with a shortened exposure time of around 0.7 ms/pixel which is by now the shortest acquisition time of commercial electron-multiplying charge-coupled device (EMCCD) as far as we know [43]... These dramatic improvements can be attributed to three factors. First, GERTs can produce really strong Raman signals, making it possible to reduce the acquisition time down to 0.7 ms/pixel. Second, Raman mapping is realized by rapid movement of the laser spot across the labels instead of mechanical movement of the stage, thus significantly shortening scanning time. The laser movement in *X* and *Y* directions is controlled by two galvo mirrors, which rotate fast around two orthogonal axes, respectively. Third, the detector processes the collected data line by line in the SWIFT mode rather than pixel by pixel, thus greatly reducing transmission and processing time of data. It should be noted that the readout speed of Raman-based PUF labels at present still lags behind that of some other PUF labels, such as silicon PUFs [56], and is not enough yet to meet the requirement for the manufacturer's registration. Nevertheless, the scanning speed of the lab-based confocal Raman system will be further continuously improved for practical use with many strategies and even the Raman mapping can be potentially realized on a hand-held device in the future. Apart from the synthesis

of Raman tags with better performance, new Raman imaging modes could be introduced, such as the application of line-shaped [57, 58] or multipoint laser [59-62], direct Raman imaging with a narrow-band filter [63], or a mode, where the stage movement, light collection and data readout occur continuously and synchronously [57].”

Please also refer to the **Response to the Comment 1** of the **Reviewer 2** for some comments about the novelties in this work.

Figure R7. High-speed readout of PUF labels. (a) Schematic of configurations of Raman system used for high-speed scanning. Under the DuoScan mode, two galvo mirrors rotate along two orthogonal axes in the mirror plane, controlling movements of the laser spot along the X and the Y axis, respectively. (b) 2D (top) and 3D (bottom) plot for a PUF label ($100 \times 100 \mu\text{m}^2$) read in DuoScan mode in 6 s with a resolution of 50×50 pixels.

Reviewer 3

Comment 1: One of the major objectives of the authors is to enlarge the encoding capacity of the PUF. This is not the only concern of a PUF system. A PUF with larger encoding capacity can reduce the probability of fabricating two tags by a stochastic process and resulting in the same response. We also need to understand the distribution of the responses in the encoding capacity, otherwise, it may still be relatively easy for an attacker to fabricate another PUF that has the same

response or has a close enough response which can lead to a false positive. The authors also mentioned that the distribution of nanoparticles is not even on the whole surface, then it will be very helpful to characterize the distributions of nanoparticles in every pixel. From the statistical study of every pixel, we will know how much randomness exists in each pixel and how much security we can get from the large encoding capacity. In other words, although the encoding capacity of the proposed PUF is very large, the probability of each response is not equal, so an attacker can use the response which has the highest probability of being fabricated to guess an arbitrary response. For example, in fig 3 f (iii), most of the pixels are encoded as 1, so the best guess for an attacker will be a response with all 1s.

Response: Thanks for pointing out the concern. We agree with the reviewer very much that the ununiform distribution of the response each pixel will easily lead to false positives. In the new version, we have scanned much more PUF labels (totally 100 PUF labels) with a resolution of 50×50 pixels as the sample database for building up the robustness of our authentication system. Each label was read three times. In the previous version, as indicated in Figure R1a, the digitized labels have shown nonuniform distributions of intensity levels (many '0' in the matrices). We carefully analyze the readouts and find out that this issue is most likely induced by the compression of many pixels into a small intensity range after normalization due to several ultra-bright pixels. Thus, we adopt a new method of **Z-score** to process the readout matrices. With this Z-score method, the matrix is standardized to a set with the mean of "0" and the standard deviation of "1". Z-score helps to significantly reduce the effects of the extremely small portion of the ultra-bright pixels in the whole matrix, resulting in much more uniform distribution of different intensity levels. As a result, four intensity levels (0, 1, 2 and 3) are more uniformly distributed, as indicated in Figure R1b and Figure 3f iii (see more details from the **Response to Comment 1 of Reviewer 1**).

In fact, unless the particle concentration is small enough, PUF labels fabricated by drop-casting NP solution on substrates to produce random patterns will have the problem of incomplete randomness. It does have the possibility of a label with a higher probability. However, as the number of pixels increases, the probability of each response will decrease significantly. The labels with relatively high occurrence probability only account for a very small portion of all possible labels, so it is difficult for an attacker to forge the same labels in a random way. At the same time,

it is difficult for an attacker to produce a specific label due to its unrepeatability nature. Another way to fight against counterfeiting is that the labels manufactured for commercial use merely occupy an extremely small portion of all possible labels. Frequency can be used to approximate the representation of probability in statistics. When a large number of labels are produced, those with an ultra-high degree of similarity will not be included in the database, so it is difficult for an attacker to pass the validation by forging labels with high occurrence probability. Although this approach reduces the encoding capacity, the extremely huge theoretical encoding capacity makes the number of possible labels in the database still considerable.

Comment 2: In the authentication application, the authors need to clarify their attacker model. The authors need to specify what are the trusted components. For example, if the complete Raman system is not trusted at the verifier side, an attacker can use his own Raman system to measure the response of a legitimate tag and replay it to the verification server to pass the verification.

Response: Thanks for raising this question. We really agree with the reviewer that the attacker model plays an important role in PUF security. However, it is a common issue for all kinds of PUF labels. In fact, our article mainly demonstrates the concept of SERS-based PUF labels, so we do not make many assumptions about possible loopholes in practical applications.

Raman systems convert Raman optical signals into digital signals, and then the computer compares the digitization with existing references in the database to determine a match or non-match. During this process, the most likely vulnerability is that the digital signals uploaded to the cloud server may be replaced and repeatedly verified. However, we believe that the collected digital signals can be further algorithmically encoded and uploaded to the cloud server, where the decoding and verification process can be carried out and feed back to the terminal device. **The whole process could effectively avoid direct transportation of digital signals caused by possible attacks. This approach can solve the problem that the response for validation may be substituted.**

Comment 3: The authors improved the read-out speed of the PUF, such that it takes 6 seconds for measuring the response of a tag. Please justify the practicality of the proposed PUF when it is compared with a silicon PUF, which usually takes less than 1 millisecond to measure, and does not require a complex optical system for measurement. Please give a reference to back up the

claim that the total measurement time is 20 minutes.

Response: Thank the reviewer for pointing out the readout speed issue of the SERS tags-based PUF labels. We have to admit that the readout speed of Raman tags-based PUF labels is much slower compared to the silicon PUFs at present.

Therefore, we have added more corresponding discussion in the revision as follows.

Page 24: “It should be noted that the readout speed of Raman-based PUF labels at present still lags behind that of some other PUF labels, such as silicon PUFs [56], and is not enough yet to meet the requirement for the manufacturer’s registration. Nevertheless, the scanning speed of the lab-based confocal Raman system will be further continuously improved for practical use with many strategies and even the Raman mapping can be potentially realized on a hand-held device in the future. Apart from the synthesis of Raman tags with better performance, new Raman imaging modes could be introduced, such as the application of line-shaped [57, 58] or multipoint laser [59-62], direct Raman imaging with a narrow-band filter [63], or a mode, where the stage movement, light collection and data readout occur continuously and synchronously [57].”

However, it should be also noted that silicon PUFs are more interesting in terms of manufacturing cost and readiness in integrating with computing and variations to produce a novel signature for each IC. According to the different sources of variation, there are three major silicon PUFs available. They are memory-based PUFs, analog electronic PUFs, and delay-based PUFs. Silicon PUFs, however, suffer from several draw backs including low entropy (DRAM), high power consumption (SRAM), and area inefficiency due to the involvement of additional tamper detection circuitry. Further, Si-PUFs are also susceptible to environmental variations, aging, side channel attacks, and hardware trojans, and they also lack reconfigurability, which make them weak PUF (*Communications Physics*, 2019, 2, 39). Moreover, the recent decline of the Si technology owing to the slowdown of Moore’s law of scaling have facilitated the growth in nanotech PUFs based on novel nanomaterials, such as randomly nanoparticles, self-assembled carbon nanotubes (CNTs), sub-lithographic random network of metal wires, block copolymers, and memristive crossbar arrays. Optical PUFs offer better security solution since they are non-volatile, low power, and, to a limited extent, reconfigurable. Reverse engineering is difficult for optical PUF and they are regarded as strong PUF.

We have also added the reference (*ACS Appl. Mater. Inter.* 2017, 9, 3995) to back up the claim

that the total measurement time is 20 minutes in the revision.

Comment 4: The authors claimed that to use this tag in practice (improve the true positive rate and reduce the false positive rate), every tag needs its own optimum digitization threshold which will be stored in the verification server as well. Please elaborate on the process of finding this optimum threshold, and how long this process takes. Also, please comment on how this optimum threshold will affect the similarity index different PUFs and the same PUFs, and therefore effectively shrink the encoding capacity.

Response: Thank the reviewer for pointing out this question. After careful consideration and taking the suggestions of the reviewers, we finally believe that a common threshold is needed for digitization of PUF labels of the same type. In the new version, we have scanned much more labels (totally 100 PUF labels) to determine the threshold. Each label has been measured three times. The 70 PUF labels were used to find the threshold with the help of global optimization algorithm and the other 30 PUF labels were used to verify that the threshold is reasonable. The optimization function was set as

$$\max \sum_{i=1}^N \sum_{j=i+1}^N \text{sign}(x_i, x_j) \times \text{Accu}(x_i, x_j)$$
$$\text{sign}(x_i, x_j) = \begin{cases} +1, & \text{if } x_i \text{ and } x_j \text{ are responses of the same label} \\ -1, & \text{if } x_i \text{ and } x_j \text{ are responses of the different labels} \end{cases}$$
$$\text{Accu}(x_i, x_j) = \frac{\text{The pixel number with the same value between } x_i \text{ and } x_j}{\text{pixel number}}$$

where N is the number of the response, here $N = 70 \times 3 = 210$. The optimization process aims to find the most appropriate threshold for digitization, which should result in not only relatively high reproducibility for the same labels but also significant disparity between different ones. It takes about 2 min to calculate the best threshold for the 100 PUF labels, and it is a one-off time consumption which means that the same calculation is not necessary for newly-fabricated labels.

The digitized matrices then go through pairwise comparison to get similarity indexes I of the same labels (300 sample capacity) and those of different ones (21735 sample capacity). We have plotted the distribution histograms of I from the 100 PUF labels (see Figure R2 and Figure 5g, h). From the histogram, we can see similarity indexes of the same PUF labels can be well separated from those of different labels, which means that the optimized threshold would result in no false

positives. Meanwhile, the smallest similarity indexes for the same labels are around 90% and 75% for binary and quaternary encoding, respectively. It indicates that the threshold can lead to relatively high reproducibility of the same labels, which will increase the real encoding capacity of the PUF labels when a relatively high error margin is introduced for validation. Please see more details from the **Response to Comment 1 of Reviewer 1**.

Reviewers' comments:

Reviewer #1 (Remarks to the Author):

Dear Authors

I must commend you. You've made a strong effort to address all the comments made on your manuscript.

I cannot find a sentence that I object to, and by introducing a new figure of merit I can support the publication of this manuscript. A manuscript describing the PUF with highest encoding capacity I've seen.

Good job

Thomas Just Sørensen

PS. A language editor will be able to make the manuscript even better.

Reviewer #2 (Remarks to the Author):

The revision has addressed my earlier comments. Thus I recommend its publication in nature communication.

Reviewer #3 (Remarks to the Author):

I am glad to see most of my concerns are appropriately addressed. Notably, the authors introduced new digitalization methods and more empirical study to improve the distribution of the responses and the digitalization process.

I have one comment remaining. The authors responded to one of my comments regarding the attacker model in the response letter, but the authors still did not add the attacker model in the paper. I believe that a clear and precise attacker model is still needed for the authentication protocol. Note that, in a security paper, the attacker model is necessary for the readers to understand the materials of the paper. It will not weaken the paper by stating that something needs to be trusted; it actually makes the paper clearer and better.

Response to reviewers' comments

Reviewer 1

Comment 1: I must commend you. You've made a strong effort to address all the comments made on your manuscript. I cannot find a sentence that I object to, and by introducing a new figure of merit I can support the publication of this manuscript. A manuscript describing the PUF with highest encoding capacity I've seen. Good job.

Response: Thanks so much for your recognition and putting so many efforts in reviewing our manuscript.

Reviewer 2

Comment 1: The revision has addressed my earlier comments. Thus I recommend its publication in nature communication.

Response: Thank you so much for the recognition and helpful comments and putting so many efforts in reviewing our manuscript.

Reviewer 3

Comment 1: I am glad to see most of my concerns are appropriately addressed. Notably, the authors introduced new digitalization methods and more empirical study to improve the distribution of the responses and the digitalization process.

Response: Thank you very much for your recognition of our manuscript, especially the revision made to the digitization process.

Comment 2: I have one comment remaining. The authors responded to one of my comments regarding the attacker model in the response letter, but the authors still did not add the attacker model in the paper. I believe that a clear and precise attacker model is still needed for the authentication protocol. Note that, in a security paper, the attacker model is necessary for the readers to understand the materials of the paper. It will not weaken the paper by stating that something needs to be trusted; it actually makes the paper clearer and better.

Response: Thank you for pointing out lack of attacker model in the manuscript. We really agree with the reviewer that the attacker model is indispensable in security paper. **Therefore, we have made assumptions about possible loopholes of our PUF system and given possible solutions, and the corresponding discussion has been added in the revision as follows.**

Page 22: "... During the authentication process, the most likely existing vulnerability is that the digitized signals uploaded to the cloud server may be replaced by an attacker, who can scan and get the digital signal of one genuine PUF label and repeatedly apply it to replacing uploaded signals of fake labels. However, we believe that the digitized data can be further algorithmically encoded and then sent to the cloud server, where the decoding and verification process is carried out followed by the feedback to the terminal device. With the help of an effective coding algorithm which can be renewed frequently, the PUF system could effectively avoid direct transmission of digitized signals, thus solving the problem of data substitution by attackers."

In addition, we have updated Figure 4 and Figure S5-9 for more uniform distributions of digitized matrices of PUF labels. Also, Roman numerals have been removed from all figures and the corresponding discussions of the manuscript in the revision.